# Van der Waals β-Ga₂O₃ thin films on polycrystalline diamond substrates

Jing Ning [1,2,3,4] ✉, Zhichun Yang[1,2,4], Haidi Wu[1,2], Xinmeng Dong[1,2], Yaning Zhang[1,2], Yufei Chen[1,2], Xinbo Zhang[1,2], Dong Wang[1,2,3], Yue Hao[1,2,3] & Jincheng Zhang[1,2,3] ✉

The self-heating effect in wide bandgap semiconductor devices makes epitaxial Ga₂O₃ on diamond substrates crucial for thermal management. However, the lack of wafer-scale single-crystal diamond and severe lattice mismatch limit its industrial application. This study presents van der Waals β-Ga₂O₃ (VdW-β-Ga₂O₃) grown on high-thermal-conductivity polycrystalline diamond. VdW forces modify the coupling state between the single-crystal thin film and polycrystalline substrate. Tunable growth of ($\bar{2}$01) VdW-β-Ga₂O₃ is achieved by leveraging the mismatch between graphene and the oxygen surface densities of varying crystal orientations and their oxygen-partial-pressure dependence. The 350 nm thick, high-crystallinity films exhibit a smallest rocking curve FWHM value of 0.18° and a root mean square roughness of 6.71 nm. Graphene alleviated interfacial thermal expansion stress; β-Ga₂O₃/diamond interface exhibits an ultralow thermal boundary resistance of 2.82 m²·K/GW. Photodetectors exhibit a photo-to-dark current ratio of 10⁶ and a responsivity of 210 A/W, confirming the strategy's practicality and technological significance.

As a wide bandgap semiconductor, gallium oxide (Ga₂O₃) has attracted extensive research interest due to its rich material properties[1,2], making it an ideal candidate for next-generation high-power electronic devices[3] and ultraviolet optoelectronic applications[4]. The β-phase Ga₂O₃ exhibits a breakdown electric field 27 times higher than silicon and 2.4 times greater than GaN, demonstrating exceptional suitability for ultra-high voltage MOSFETs and Schottky diodes[5]. With a Baliga's figure of merit (BFOM) 3000 times superior to silicon and 4 times better than GaN[6], β-Ga₂O₃-based devices achieve significantly lower conduction loss and higher efficiency at equivalent voltage ratings. Owing to its outstanding properties including high breakdown field strength, low energy loss, excellent radiation hardness[7], thermal stability[8], and chemical stability[9], β-Ga₂O₃ outperforms conventional wide-bandgap semiconductors by orders of magnitude in power electronics applications.

Despite the notable performance advantages of Ga₂O₃, it presents considerable drawbacks. One critical limitation is its relatively low thermal conductivity, approximately 10–30 W m⁻¹ K⁻¹, which is only one-sixth that of diamond[10,11]. This weakness poses considerable challenges for high-power semiconductor device materials. As miniaturization progresses and power levels rise, the reliability and stability of Ga₂O₃-based power devices encounter formidable challenges. For instance, the thermal accumulation effect rapidly intensifies with rising power density, leading to performance degradation and preventing the full utilization of Ga₂O₃'s high-power potential. Consequently, thermal management has emerged as one of the primary technical bottlenecks restricting the development and widespread application of Ga₂O₃-based power devices[12,13]. Enhancing thermal transport from the proximal junction heat region of Ga₂O₃-based devices is a promising solution. Diamond, with its ultra-high thermal conductivity, has gradually emerged as the preferred heat sink material for Ga₂O₃ devices[14,15]. Previous attempts to integrate Ga₂O₃ with diamond have employed low-temperature direct bonding[16] and ion-cutting

[1]The State Key Laboratory of Wide-Bandgap Semiconductor Devices and Integrated Technology, Xi'an, China. [2]Shaanxi Joint Key Laboratory of Graphene, Xidian University, Xi'an, China. [3]Xidian-Wuhu Research Institute, Xidian University, Wuhu, China. [4]These authors contributed equally: Jing Ning, Zhichun Yang. ✉e-mail: ningj@xidian.edu.cn; jchzhang@xidian.edu.cn

techniques[17]. However, these methods encounter challenges, including the formation of interfacial amorphous layers at high temperatures and irradiation-induced deep-level defects. Consequently, the epitaxial growth of $Ga_2O_3$ on diamond substrates has emerged as a promising research direction[18,19]. To date, researchers have successfully epitaxially grown β-$Ga_2O_3$ with a full width at half maximum (FWHM) between 3° and 4° on (111) single-crystal diamond substrates[20] and κ-$Ga_2O_3$ on polycrystalline diamond substrates for photodetector applications[21]. However, notable challenges persist, including the lack of wafer-scale single-crystal diamond substrates and issues such as low crystallinity and high interfacial defects associated with conventional epitaxy. Virtually no progress has been achieved globally in the epitaxial growth of single-crystalline β-$Ga_2O_3$ on polycrystalline diamond substrates.

The core challenge associated with growing $Ga_2O_3$ on polycrystalline diamond substrates stems from the disordered orientation of diamond grains, making the mitigation of substrate influence critical[22–24]. Traditional epitaxy, which relies on covalent or ionic bonding, induces substantial stress, dislocations, and defects when lattice matching is poor—an issue exacerbated by the complex and asymmetric lattice structure of $Ga_2O_3$. In this work, we introduce the innovative concept of van der Waals β-$Ga_2O_3$ (VdW-β-$Ga_2O_3$), wherein materials are bound by weak VdW forces[25–29]. By employing a two-dimensional (2D) material as a lattice shielding layer, we achieve flexible control of interface coupling between the material and the substrate[30–33]. Using mist chemical vapor deposition (mist-CVD), we epitaxially grew highly oriented, pure-phase VdW-β-$Ga_2O_3$ films on polycrystalline diamond substrates. This non-vacuum method reduces energy costs via simple atomization/heating systems, using affordable, safe precursors without toxic byproducts. It enables scalable, large-area deposition and excels at low-temperature nucleation on inert surfaces (e.g., graphene/h-BN) through reactive intermediates (−OH/ −COOH) from partially decomposed precursors, unlike MOCVD/MBE's high-temperature/plasma requirements. Droplet-enhanced adsorption further aids uniform growth. Mist-CVD avoids wafer bonding's interfacial defects/stress or amorphous layer risks. These advantages suit both lab and industrial use[37].

## Results and discussion

### Growth and fabrication of Van der Waals β-$Ga_2O_3$

The β-$Ga_2O_3$ crystal belongs to the hexagonal system and can be represented using a hexagonal close-packed model, where gallium ions occupy the centers of the hexagons and oxygen ions are positioned at the vertices. Graphene and h-BN form planar hexagonal honeycomb structures through sp² hybridization, which theoretically enables $Ga_2O_3$ epitaxy on polycrystalline substrates. To assess the feasibility of this epitaxy, we constructed atomic models of oxygen atoms and bulk $Ga_2O_3$ adsorbed onto monolayer (ML) graphene/diamond and monolayer hexagonal boron nitride (ML-h-BN)/ diamond (Fig. 1a, c). The high adsorption energy displayed in Fig. 1b indicates effective nucleation induced by the 2D materials. Specifically, for the [100] orientation, the inclusion of 2D materials considerably increases the adsorption energy from −1.33 eV to −2.99 eV (graphene) and −3.09 eV (h-BN). A top-view analysis reveals that adsorption on hexagonal vacancies is more stable than that at grid-top positions. Figure 1d, e illustrate the adsorption energy trends of oxygen atoms and the $Ga_2O_3$ bulk material as a function of the number of graphene and h-BN layers, assuming a 3.5 Å interlayer spacing.

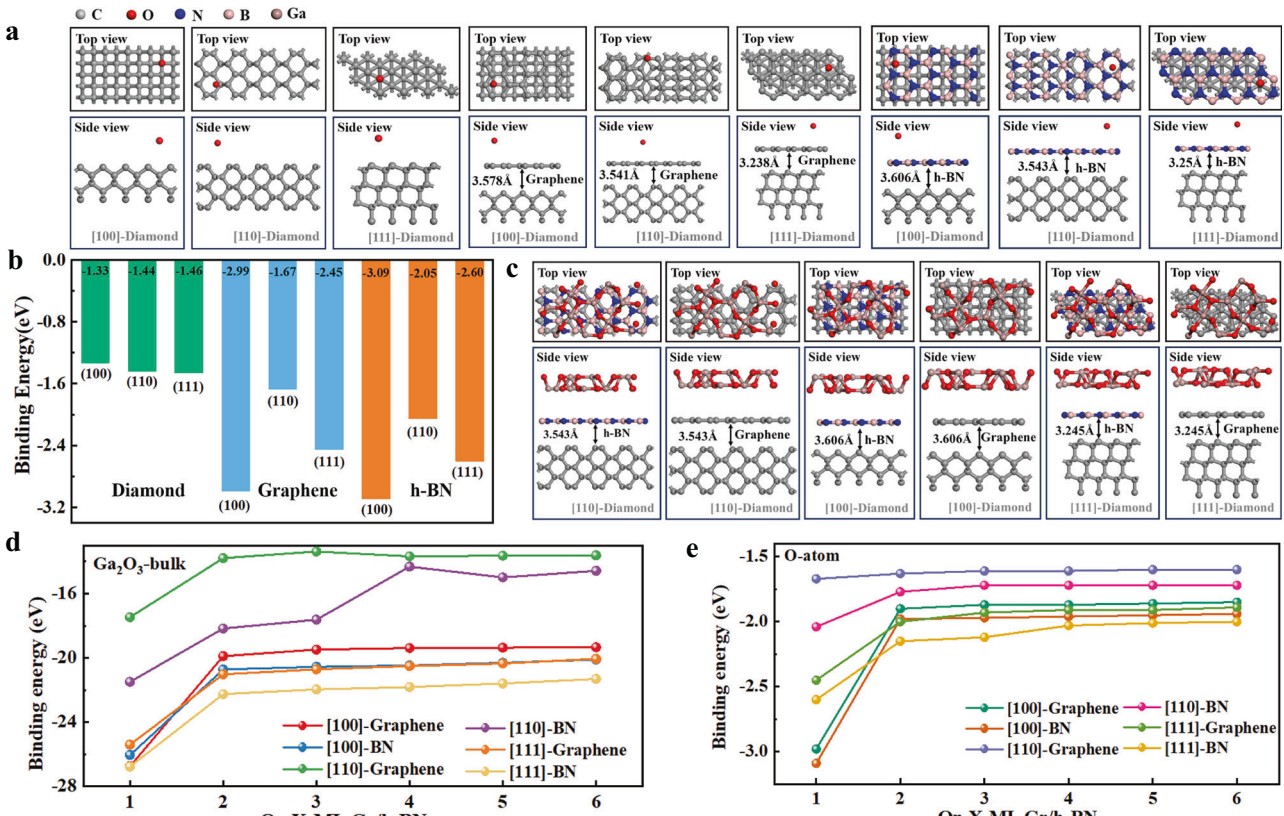

**Fig. 1 | Adsorption energy calculations of VdW-β-$Ga_2O_3$ epitaxy on polycrystalline diamond. a** Atomic structures of O atoms adsorbed on ML-graphene/ diamond, ML-h-BN/diamond, diamond substrate visualized. The orientations of the diamond are [100], [110], and [111]. Both top and side views are provided. **b** Adsorption energies of oxygen atoms for the configurations depicted in (**a**). **c** Schematics illustrating the atomic structures of epitaxial $Ga_2O_3$ bulk material with ML-h-BN and Graphene as insertion layers. **d** Adsorption energy trends for epitaxial $Ga_2O_3$ and **e** O atoms as a function of the number of graphene and h-BN layers, with the number of insertion layers ranging from 1 to 6. Source data are provided as a Source Data file.

The data indicate that the adsorption energies of both oxygen atoms and $Ga_2O_3$ are maximized when a single insertion layer is used. As the number of graphene and h-BN layers increases, the binding energy drops and gradually stabilizes. This trend suggests that the interaction between the epitaxial material and the substrate weakens as the distance increases. This is because the weak VdW forces and distant atomic interactions cannot provide stronger binding. At this point, the VdW forces of the 2D materials dominate the epitaxial film behavior. Although polycrystalline substrates consist of randomly oriented grains, the interfacial adsorption energy is primarily determined by local atomic arrangements. Our experimental characterization confirms that most exposed grain surfaces match the model configuration, validating the representativeness of calculations. Furthermore, our computational predictions demonstrate that monolayer graphene/BN can significantly enhance adsorption energy, with improvements observed across various crystal orientations. Based on these findings, monolayer graphene was selected as the intercalation layer for the experiment.

Transmission electron microscopy (TEM) was performed to examine the atomic arrangement of VdW-β-$Ga_2O_3$. The cross-sectional TEM image (Fig. 2a) clearly illustrates the interface distribution between VdW-β-$Ga_2O_3$, graphene, and polycrystalline diamond. VdW-β-$Ga_2O_3$ exhibits two distinct crystal orientations: ($\bar{2}01$) and ($\bar{4}01$), with ($\bar{2}01$) being the predominant one, characterized by a clearer and more orderly atomic arrangement. The High-resolution TEM (HR-TEM) images in Figs. 2b, c, along with the atomic structure schematics, reveal

that the carbon atom spacing in the monolayer graphene is 0.492 nm, which closely matches the theoretical oxygen atom spacing of 0.496 nm in the ($\bar{2}01$) β-$Ga_2O_3$ orientation. In contrast, Fig. 2d indicates that the oxygen atom spacing in the ($\bar{4}01$) β-$Ga_2O_3$ orientation is 0.215 nm, explaining the preferential growth of $Ga_2O_3$ in the ($\bar{2}01$) orientation. The interplanar spacings measured at the cross-section are approximately 0.59 nm and 0.22 nm, respectively (Fig. 2d). The notable difference in atomic plane spacings is attributed to the island-like growth mode of the ($\bar{2}01$)-oriented $Ga_2O_3$. During the transition from 3D island growth to 2D film coalescence, compression occurs, leading to extrusion of crystals at the coalescence sites along different directions. Figure 2e presents the top-view atomic structures of the ($\bar{2}01$) $Ga_2O_3$ and (001) graphene orientations. The oxygen atom spacings in the ($\bar{2}01$) orientation are 2.98 and 3.04 Å, while the carbon atom spacings in the (001) orientation of graphene is 2.46 and 2.84 Å. Using the formula $\Delta a/a = (a_1 - a_2) / a_1$, the lattice mismatch coefficient between β-$Ga_2O_3$ and graphene is calculated to be as low as 4.9%. Given that the theoretical threshold for the lattice mismatch rate is 6%, this low mismatch ensures structural stability. These results confirm the successful epitaxial growth of VdW-β-$Ga_2O_3$ on graphene.

To determine the optimal growth conditions for epitaxial VdW-β-$Ga_2O_3$ films, XRD and XPS characterization were performed. Figure 3a presents XRD scans of samples grown at various temperatures, revealing prominent diffraction peaks corresponding to the ($\bar{2}01$), ($\bar{4}01$), and ($\bar{2}02$) planes. The peak of the α-phase marked with black diamond, which occurs because the temperature of 700 °C remains

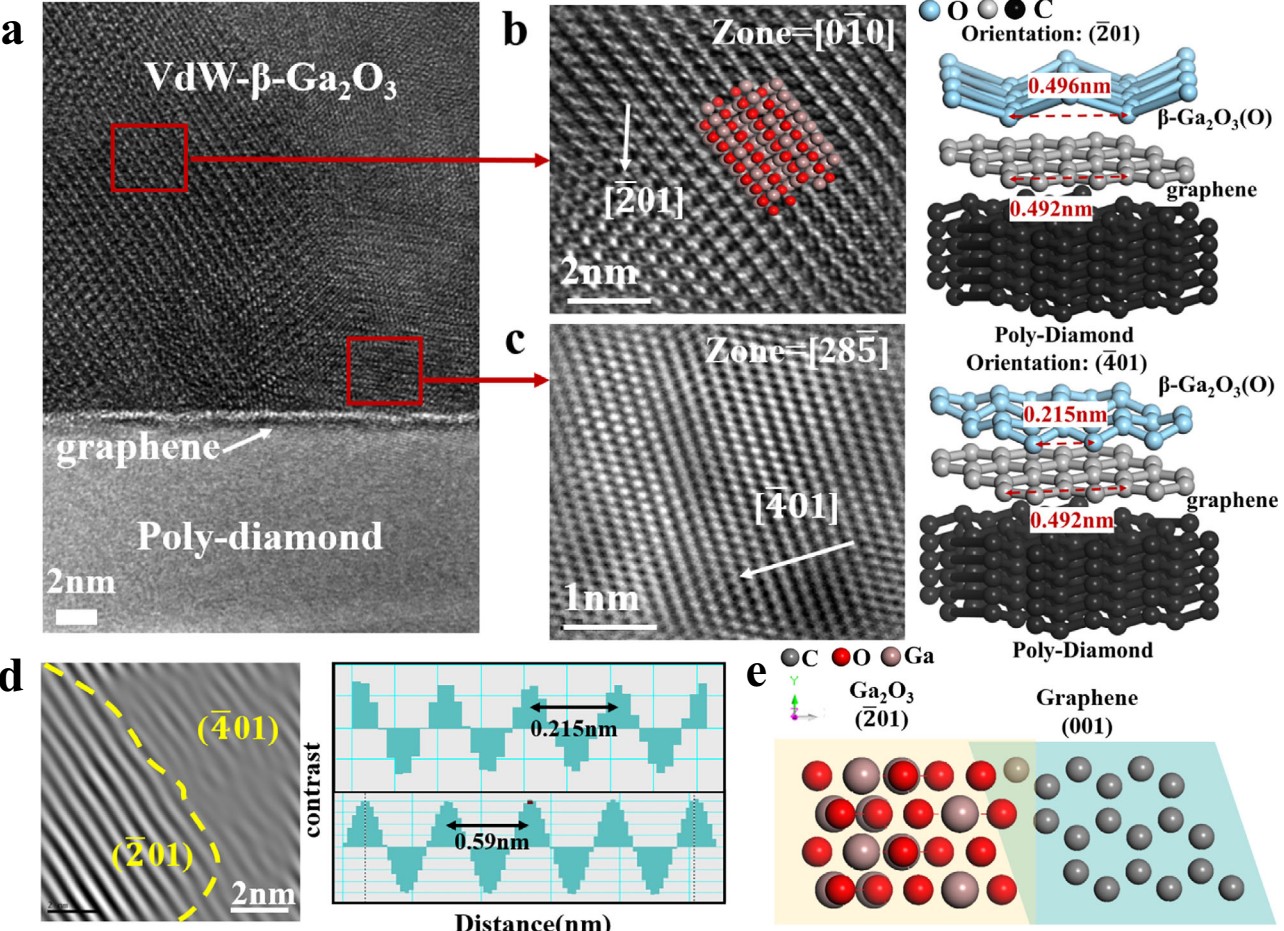

**Fig. 2 | TEM images of VdW-β-$Ga_2O_3$ on poly-diamond substrates. a** High-magnification cross-sectional TEM image of the interface between VdW-β-$Ga_2O_3$, ML graphene, and poly-diamond. **b** HR-TEM images of the ($\bar{2}01$) β-$Ga_2O_3$ and **c** ($\bar{4}01$) β-$Ga_2O_3$ crystal orientations, with corresponding atomic structure schematics. **d** Crystal interface between the ($\bar{2}01$) β-$Ga_2O_3$ and ($\bar{4}01$) β-$Ga_2O_3$ orientations, including the measured interplanar spacings. **e** Top-view atomic structures of the ($\bar{2}01$) β-$Ga_2O_3$ orientation and (001) graphene.

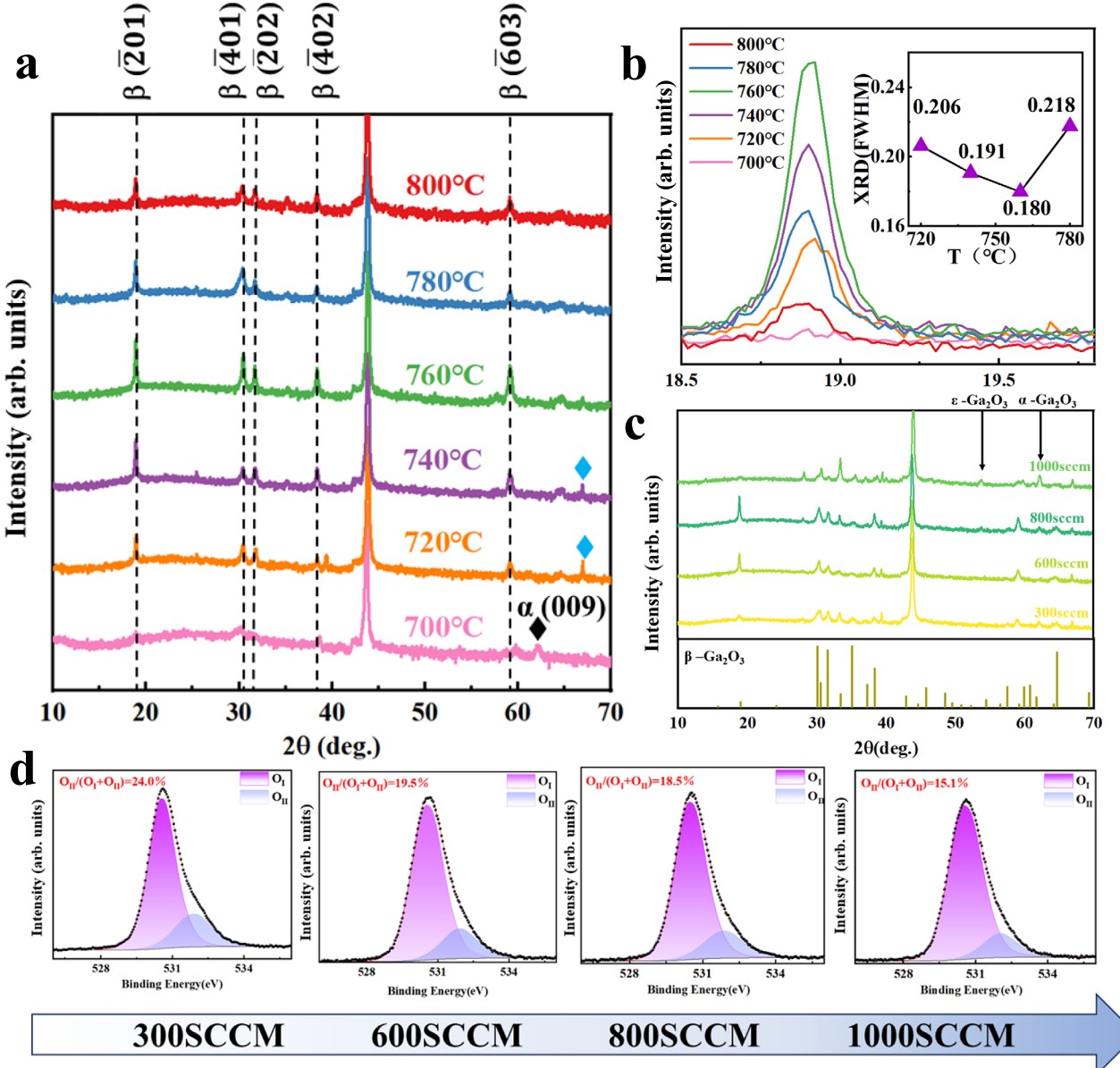

**Fig. 3 | X-ray diffraction (XRD) and X-ray photoelectron spectroscopy (XPS) spectra of VdW-β-Ga$_2$O$_3$ thin films. a** XRD spectra under different temperature conditions. **b** Amplified XRD rocking curve of the ($\bar{2}$01) VdW-β-Ga$_2$O$_3$ reflection (Inset: Trends of FWHM variations). **c** XRD spectra of VdW-β-Ga$_2$O$_3$ at various O$_2$ flow rates. **d** XPS spectra of the O1*s* peak under different O$_2$ flow rates. Source data are provided as a Source Data file.

relatively low and insufficient to meet the conditions for irreversible transformation of the metastable α-phase into the β-phase at higher temperatures. Additionally, sharp, zero-width peaks marked with blue diamonds were observed under 740 and 720 °C conditions, suggesting these "needle-like peaks" may represent instrumental noise or artifacts. Specifically, the peaks at ($\bar{2}$01), ($\bar{4}$02), and ($\bar{6}$03) indicate a consistent β-phase oriented along the <$\bar{2}$01> direction. Figure 3b presents an enlarged view of the XRD rocking curve for the ($\bar{2}$01) VdW-β-Ga$_2$O$_3$ reflection and its FWHM. The smallest rocking curve FWHM value of 0.18° occurs at 760 °C, where the 350 nm thick films (Supplementary Fig. 1) exhibit both the highest crystallinity and fewest interfacial defects. Figure 3c compares the XRD spectra at varying O$_2$ flow rates. Our rigorous sample storage protocol ensures the temporal validity of the acquired data. (Supplementary Fig. 2) Peaks corresponding to other gallium compounds appear at 1000 sccm and 300 sccm, suggesting insufficient oxygen availability at low flow rates and incomplete precursor reactions at high flow rates. XPS characterization was

conducted within the range of 0–1300 eV. Figure 3d displays the O 1*s* signal peak at approximately 530.8 eV. Gaussian fitting identifies two components: a strong peak at 530.7 eV, corresponding to oxygen ions (O$_I$) in the lattice, and a secondary peak at 531.7 eV, associated with oxygen vacancies (O$_{II}$)[38,39]. The peak area ratio of O$_{II}$/(O$_I$ + O$_{II}$) provides an estimate of the relative oxygen vacancy density in the Ga$_2$O$_3$ film. The results indicate that oxygen vacancies decrease with increasing oxygen flow rate. This is primarily attributed to the higher oxygen partial pressure at higher O$_2$ flow rates, which enhances the activity of lattice atoms in O$_2$ and promotes the incorporation of oxygen atoms. Consequently, Ga–O bond formation is strengthened, reducing the density of oxygen vacancies in the lattice.

**Mechanism of tunable growth for ($\bar{2}$01) Van der Waals β-Ga$_2$O$_3$**
To illustrate the impact of growth conditions on crystal orientation, Fig. 4a presents the three primary orientations observed in the XRD spectra of Ga$_2$O$_3$ cross-sections. The ($\bar{2}$01) plane exhibits the highest

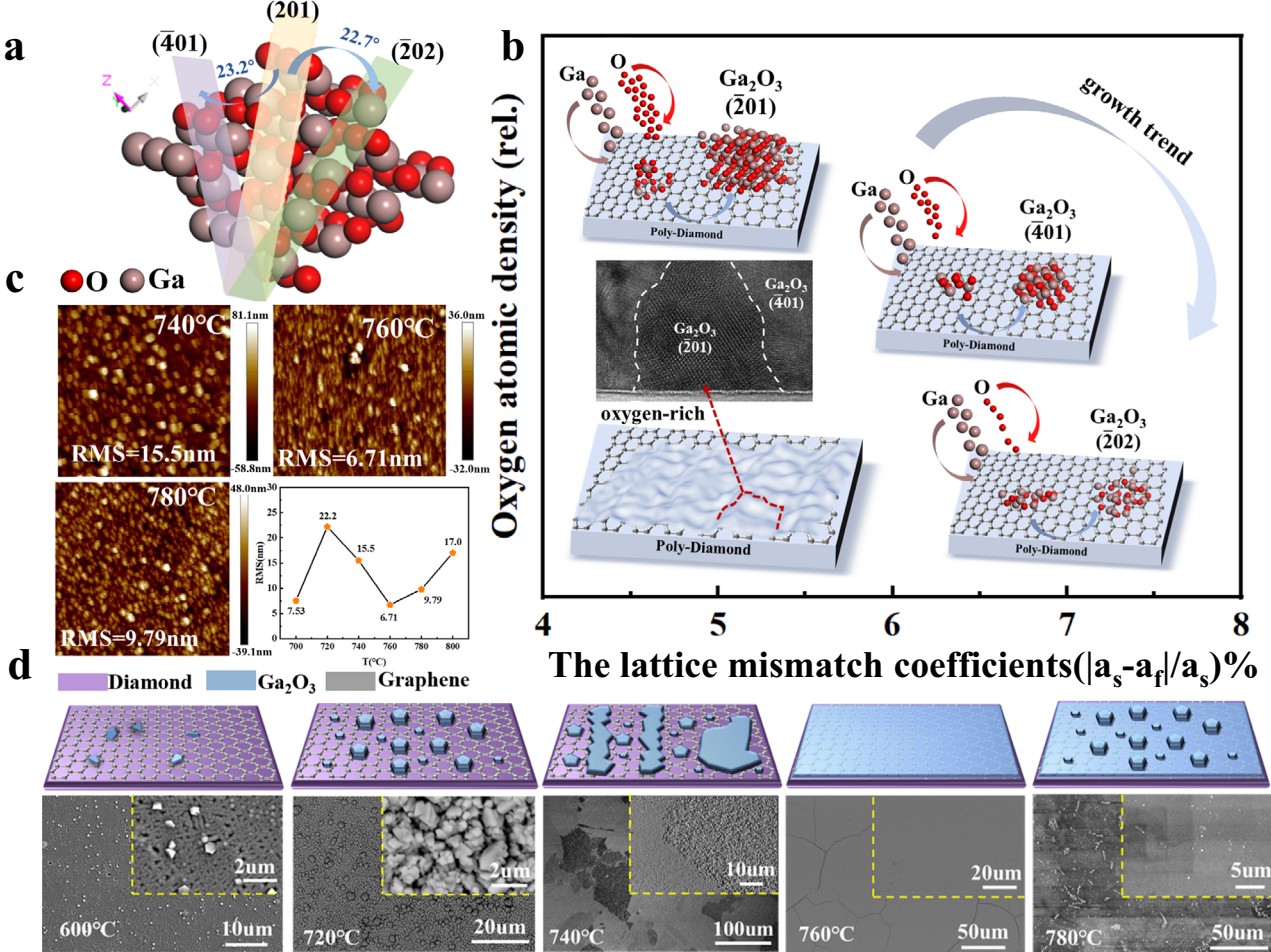

**Fig. 4 | VdW-β-Ga₂O₃ on polycrystalline diamond. a** Schematic illustrating atoms at different crystal orientation cutting surface positions within the Ga₂O₃ crystal structure. **b** Influence of lattice mismatch (aₛ and aғ represent the lattice constant of the substrate and epitaxial layer, respectively.) and oxygen atom density on the crystalline orientation growth trend. (Inset: schematic of the preferential growth of the ($\bar{2}$01) orientation and a TEM image.) Controlled growth of ($\bar{2}$01) orientation can be achieved with increasing oxygen partial pressure and decreasing lattice mismatch coefficient. **c** AFM surface morphology images at 740, 760, and 780 °C, showing the dependence of sample roughness on deposition temperature. (Source data are provided as a Source Data file.) **d** Regular and local magnified SEM images and corresponding schematics of samples grown at 600–800 °C. (This figure displays representative results from four independent experiments).

oxygen atom density, whereas the other two planes, rotated by 23.2° and 22.7° along the X-axis, exhibit lower oxygen densities. This suggests that the O₂ flow rate can influence the tilting of these planes. The lattice mismatch coefficients, calculated as $\Delta a/a = |a_f − a_s|/a_s$ %, reveal that the ($\bar{2}$01) plane exhibits the best alignment with the substrate, with a mismatch of 4.9%, while the tilted planes display a mismatch of 7%. This is because the top view of the atomic arrangement in the ($\bar{2}$01) orientation forms a hexagonal pattern similar to that of planar graphene, which explains the preferred orientation observed in Fig. 3a. Collectively, these results indicate the influence of oxygen atom density and lattice mismatch on crystal growth orientation (Fig. 4b). The growth trend of β-Ga₂O₃ further suggests that as oxygen atom density decreases, the lattice mismatch coefficient increases, favoring the growth of the ($\bar{2}$01) crystal orientation. During the island-like growth of ($\bar{2}$01)-oriented crystals, reduced oxygen atom contacts areas and lateral coalescence lead to slight tilting of the crystal planes (by 23.2° or 22.7°), shifting the orientations to ($\bar{4}$01) and ($\bar{2}$02). Figure 4b illustrates a schematic of the island-like growth and lateral coalescence of ($\bar{2}$01)-oriented crystals, alongside a TEM cross-sectional image of the film. The distinct and orderly atomic arrangement in the ($\bar{2}$01) region confirms the validity of the oxygen-lattice co-modulation model.

Due to the influence of this polycrystalline substrate, the wet-transferred graphene shows varying degrees of wrinkles with an RMS

roughness of about 0.51 nm. The surface morphology of both diamond and graphene was characterized by AFM and SEM, as shown in Supplementary Fig. 3. Atomic force microscopy (AFM) images of the samples at varying temperatures are illustrated in Fig. 4c. The root mean square roughness initially decreases and then increases with temperature, explaining the periodic island-layer-island growth mode for the films. Combined with the TEM results and the schematic in Fig. 4d, these results suggest that at low temperatures, only a few hexagonal crystal grains adhere to the graphene surface. As the temperature increases, the grains coalesce into a smooth film, reducing surface roughness. Further temperature increases initiate a new growth cycle, causing the roughness to rise again. Distinct grain boundaries are observed at a growth temperature of 760 °C. These boundaries may be associated with defects present in the transferred graphene, the release of thermal stress after growth, or the gas flow rate during deposition. Therefore, selecting an appropriate carrier gas flow rate during growth is critical, as demonstrated in Supplementary Fig. 4.

### Thermal management and applications of thin films
To investigate the role of graphene in stress coupling between the thin film and the substrate, we conducted in situ temperature-dependent Raman measurements of β-Ga₂O₃ grown on different substrates. In

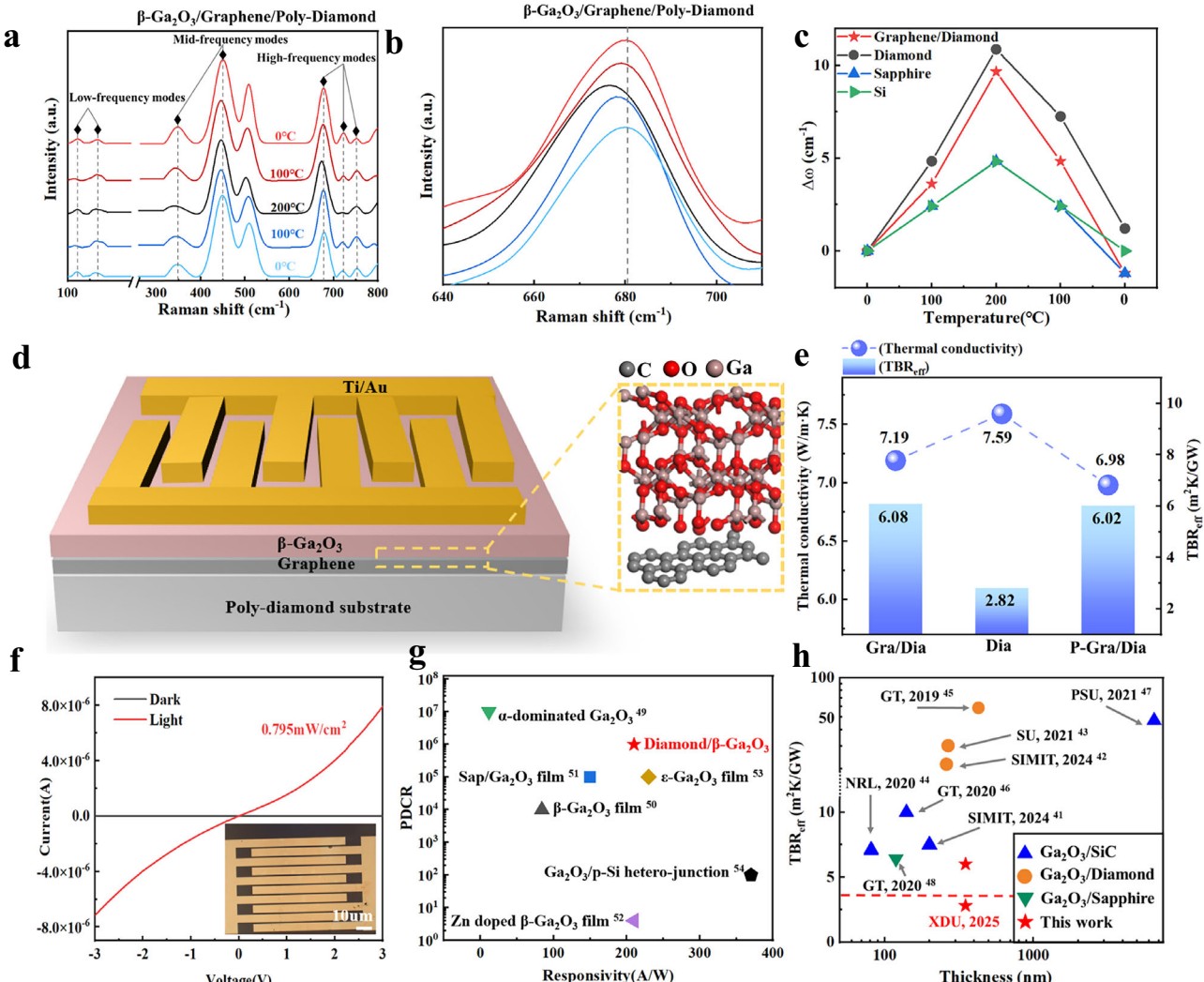

**Fig. 5 | Thermal dissipation characterization of VdW-β-Ga₂O₃ on polycrystalline diamond. a** Typical Raman spectra of β-Ga₂O₃ epitaxially grown on ML-graphene/polycrystalline diamond. **b** Local magnified Raman spectra. **c** Temperature-dependent Raman shift (Δω) of VdW-β-Ga₂O₃ on different substrate materials, extracted from in-situ Raman measurements. **d** Schematic of the photodetector structure and heat dissipation pathways. **e** Thermal conductivity of Ga₂O₃ films and interfacial thermal boundary resistance (TBR_eff) for sample1, 2 and 3. **f** Photocurrent and dark current of the photodetector under 0.795 mW/cm² illumination. (Inset: optical microscope image of interdigitated electrodes). **g** Comparison of the PDCR and responsivity of this photodetector against the performance metrics of other devices during the same period. **h** Benchmark comparison of TBR_eff values across different interface engineering approaches. (GT Georgia Institute of Technology, SIMIT Shanghai Institute of Microsystem and Information Technology, SU Stanford University, NRL U. S. Naval Research Laboratory, PSU The Pennsylvania State University) Source data are provided as a Source Data file.

principle, when β-Ga₂O₃ and the substrate exhibit similar interlayer coupling, the Raman frequency shift (Δω/ΔT) of β-Ga₂O₃ will be identical to that of the substrate owing to the equal compressive strain variation across both materials (Δε/ΔT). Figure 5a illustrates the temperature-dependent Raman spectra of β-Ga₂O₃ grown on ML-graphene/polycrystalline diamond. The high-frequency modes (at approximately 770–500 cm⁻¹) display peaks at 752 cm⁻¹ and 751 cm⁻¹, corresponding to the stretching and bending of GaO₄ tetrahedra. The mid-frequency modes (at approximately 480 to 310 cm⁻¹) exhibit peaks at 415, 344, and 453 cm⁻¹, attributed to the deformation of Ga₂O₆ octahedra. Finally, the low-frequency modes (below 200 cm⁻¹) present peaks at 188.3, 168, and 121 cm⁻¹, associated with the vibrational and translational modes of GaO₄ tetrahedra and Ga₂O₆ octahedral chains in β-Ga₂O₃[40].

After temperature cycling, the Raman peaks return to their original positions at the same temperatures, confirming that the temperature cycling process is non-destructive. As the temperature increases from 0 °C to 200 °C, the Raman peaks exhibit a red shift.

(Fig. 5b) During this temperature cycling process, the Δω values for various substrates change distinctly with temperature. Figure 5c compares the in-situ Raman spectra of β-Ga₂O₃ grown on various substrates as a function of temperature. The larger Δω/ΔT of β-Ga₂O₃ grown on the diamond substrate indicates a strong VdW coupling state and significant lattice stress release. This effect arises owing to the large thermal expansion coefficient difference between VdW-β-Ga₂O₃ and diamond. The largest Δω occurred for β-Ga₂O₃ on diamond at 100 °C, indicating the highest in-plane tensile stress, which was slightly alleviated by the graphene interlayer. The smaller Δω difference between sapphire and Si substrates (Δω_{Si,100 °C} = 2.418 cm⁻¹ vs Δω_{Sap,100 °C} = 2.413 cm⁻¹) reflects their closer thermal expansion coefficients to β-Ga₂O₃.(Supplementary Fig. 5) When the temperature was increased to 200 °C, the Δω of β-Ga₂O₃ on polycrystalline diamond continued to rise to 10.8 cm⁻¹, yielding a calculated temperature coefficient of peak shift of approximately −0.05 cm⁻¹/K over the heating range. In contrast, β-Ga₂O₃ on both Si and sapphire substrates exhibited a smaller temperature coefficient of −0.024 cm⁻¹/K, which

can be attributed to their relatively smaller thermal expansion coefficient mismatches. The slight blue shift observed at the final 0 °C measurement for β-Ga$_2$O$_3$ on Gra/Dia, Si, and Sapphire substrates is likely attributable to enhanced interfacial bonding strength induced by high-temperature treatment, which generates localized compressive stress and consequently increases phonon vibrational frequencies. These results highlight the effective role of graphene in stress release.

Here, we systematically measured the thermal conductivity of Ga$_2$O$_3$ films and the effective thermal boundary resistance (TBR$_{eff}$) at Ga$_2$O$_3$/Diamond interfaces using time-domain thermoreflectance (TDTR) for three samples: Sample 1(Graphene/poly-diamond, Gra/Dia), Sample 2(bare poly-diamond, Dia), and Sample 3(O$_2$ plasma-treated Graphene/poly-diamond, P-Gra/Dia, 50 W for 30 s). All samples underwent identical β-Ga$_2$O$_3$ epitaxy conditions. As shown in Fig. 5e, Sample 2 exhibited the lowest effective thermal boundary resistance of 2.82 m$^2$K/GW, while Sample 1 and Sample 3 showed higher values of 6.08 m$^2$K/GW and 6.02 m$^2$K/GW, respectively. This discrepancy arises from the limited phonon transport across vdW interfaces, governed by reduced actual contact area and weak adhesion energy. Although the plasma-treated graphene introduced sparse covalent bonds (marginally lowering TBR$_{eff}$), the random nucleation induced by nonuniform plasma treatment degraded film crystallinity, resulting in slightly lower Ga$_2$O$_3$ thermal conductivity. Consequently, the overall trend followed: TC(P-Gra/Dia) < TC(Gra/Dia) < TC(Dia). As demonstrated by Zheng et al., the thermal conductivity of Ga$_2$O$_3$ with specific orientations increases with thickness[11]. In our experiments, the ($\bar{2}$01) plane rocking curve showed a minimum FWHM of 0.18° for the 350-nm-thick film, corresponding to a thermal conductivity of 7.19 W/m·K. Furthermore, thinner films display more pronounced phonon boundary scattering (consistent with the Callaway model), leading to reduced effective thermal conductivity. As presented in Fig. 5h[17,18,41–46], our results demonstrate a TBR$_{eff}$ that is one order of magnitude lower than previous bonding and interlayer approaches, which is attributed to the ultraclean interface processing and optimized graphene transfer enabling atomically flat interfaces. This breakthrough fulfills the thermal management requirements for kW-class power devices.

To comprehensively evaluate material performance, metals were deposited and etched onto the sample. Figure 5d illustrates the structure of the photodetector and its corresponding heat dissipation pathways. The basic parameters of the photodetector were subsequently measured. The dark current and photocurrent of the device were measured under bias voltages ranging from −3 V to 3 V, with a light power density of 0.795 mW/cm². Figure 5f illustrates the variation in the dark and photocurrents with voltage. Specifically, the dark current (I$_{dark}$) is $5.83 \times 10^{-6}$ µA, while the photocurrent (I$_{light}$) is 7.92 µA, yielding a photo-dark current ratio (PDCR) of approximately six orders of magnitude. The inset displays an optical microscope image of the interdigitated electrodes, with an effective illuminated area of $6 \times 10^{-4}$ cm². Using the formula for responsivity R = I$_{ph}$/P$_{ill}$, we obtain an R value of 210 A/W. Figure 5g[47–52] compares the photodetector's PDCR and responsivity with those of other devices recorded during the same period. The results indicate that the device demonstrates superior PDCR performance. Furthermore, under similar PDCR values, its higher responsivity confirms the practicality and high quality of the VdW-β-Ga$_2$O$_3$ film used in this study.

We further extracted the conductivity variations of the fabricated photodetector under high optical frequencies, as shown in Supplementary Fig. 6. The results demonstrate negligible conductivity changes with increasing light frequency, indicating efficient carrier extraction and well-passivated material defects/interface states that minimize the impact of slow recombination on high-frequency response. Although high-frequency illumination may induce localized Joule heating, the stable conductivity confirms effective thermal management in the device design, with no observable interface degradation due to temperature rise. Additionally, we measured the photocurrent response under 1 Hz optical pulse illumination over an extended duration of 2000 seconds. The device exhibits remarkable robustness and reproducibility, with response times of $\tau_1 = 54$ ms (rise) and $\tau_2 = 4$ ms (decay). Collectively, these results demonstrate outstanding long-term stability of the photodetector. The scalability of the proposed methodology for practical semiconductor applications is currently constrained by two key factors: The limited availability of large-area wafer-scale single-crystal diamond substrates, and challenges in maintaining monolayer graphene uniformity across full wafer dimensions. Overcoming these scalability bottlenecks represents a critical pathway for industrial implementation in future development.

In summary, this study proposes the innovative use of a 2D material interlayer to modify the coupling state between a single-crystal thin film and polycrystalline substrate, thereby improving the formation of the Ga$_2$O$_3$ nucleation layer. Adsorption Locator module calculations indicate that the 2D material interlayer enhances the nucleation probability of Ga$_2$O$_3$, facilitating the formation and growth of epitaxial films. Based on these computational findings, monolayer graphene—a cost-effective and stability material—was selected as the interlayer. Tunable growth of ($\bar{2}$01) VdW-β-Ga$_2$O$_3$ was achieved by controlling the lattice mismatch between graphene and the oxygen densities of different crystal orientations. The surface morphology evolution and growth mechanisms of the epitaxial Ga$_2$O$_3$ films were analyzed. The results reveal that Ga$_2$O$_3$ films exhibit an island growth mode, transitioning to a continuous film upon nucleation. The growth rate increases with the oxygen (O$_2$) carrier gas flow rate, while the oxygen vacancy concentration decreases. The high-crystallinity films grown at 760 °C under 600 sccm O$_2$ carrier gas exhibited the smallest rocking curve FWHM of 0.18° and an RMS roughness of 6.71 nm at 350 nm thickness. Temperature-dependent Raman measurements of β-Ga$_2$O$_3$ films on different substrates confirmed significant lattice thermal expansion stress release between β-Ga$_2$O$_3$ and the diamond substrate, enabled by the 2D interlayer. TDTR measurements reveal an ultralow TBR$_{eff}$ of 2.82 m$^2$K/GW at the β-Ga$_2$O$_3$/diamond interface, representing a one-order-of-magnitude reduction compared with state-of-the-art reported values. Finally, a photodetector was fabricated using the highly oriented pure-phase film. The device exhibited a high PDCR of $10^6$ and a responsivity ($R$) of 210 A/W, surpassing similar metrics reported in recent studies. These results confirm the high quality, crystallinity, and uniformity of the material produced using the proposed method. This study overcomes the bottleneck of achieving direct epitaxial growth of Ga$_2$O$_3$ films on diamond substrates.

## Methods

### Graphene preparation and transfer

A 25-µm-thick Alfa Aesar copper foil was first subjected to electrochemical polishing. The polished foil was folded into a pouch of suitable size using tweezers and placed into a quartz boat. The quartz boat containing the foil pouch was positioned at the center of the heating zone within a tube furnace (Xiamen Xincheng CVD equipment). The system was maintained under a flow of 40 sccm CH$_4$ and 40 sccm H$_2$ for 40 min, followed by 10 sccm CH$_4$ and 20 sccm H$_2$ for 20 min, resulting in the formation of ML graphene fully encapsulated within the copper foil. Following CVD growth, we employed methyl methacrylate (MMA) as a transfer support layer. The copper substrate was etched using an ammonium persulfate ((NH$_4$)$_2$S$_2$O$_8$) solution, followed by multiple rinses with deionized water to completely remove residual etchant, thereby minimizing interfacial hydroxyl (-OH) group formation and its impact on material properties[53]. Prior to transferring graphene onto the polycrystalline diamond substrate, we performed ultrasonic cleaning of the diamond substrate using acetone, alcohol, and deionized water sequentially. Additionally, the substrate was immersed in dilute HF solution for five minutes to eliminate surface oxides, ensuring an atomically clean interface. We utilized polycrystalline diamond substrates from Element Six's TM180 series.

According to the manufacturer's specifications (Element Six, *Diamond Handbook 2024*), these substrates demonstrate: Thermal conductivity >1800 W/m K at 300 K, Thermal conductivity >1500 W/m K at 425 K, Polished surface roughness (Ra) < 20 nm (Supplementary Table 1).

## Mist-CVD epitaxial growth of $Ga_2O_3$

To determine the optimal conditions for the growth of β-$Ga_2O_3$ thin films on polycrystalline diamond substrates, various parameters, including temperature, carrier gas flow rate, precursor solution concentration, and HCl concentration, were investigated. A precursor solution (25–35 mL) was poured into a nebulizer, and the quartz boat loaded with graphene/diamond substrates was placed into the quartz tube. The temperature was raised to 700–800 °C under an Ar atmosphere at 2000 sccm. Once the target temperature was reached, the ultrasonic nebulizer (1.7 MHz) was activated, delivering droplets of the precursor solution onto the graphene/diamond substrates using $O_2$ as the carrier gas (300–1000 sccm) and Ar as the diluting gas (3000 sccm) for 30–40 min. The optimal $O_2$ flow rate was selected based on actual experimental data of material oxygen vacancies and comprehensive RMS values. The working principle utilizes the atomization characteristics of the ultrasonic nebulizer to convert the solution into small droplets, which are then transported into the heated reaction chamber by oxygen carrier gas. Due to the Leidenfrost effect, the droplets can float and migrate to the surface of the heated substrate. When droplets land on the high-temperature surface, the portion contacting the substrate vaporizes under heat, forming a vapor layer between them[54]. This layer prevents direct contact, and under the combined action of gravity and the vapor layer, the droplets remain suspended on the surface until complete vaporization, ultimately forming a thin film[55]. After the growth process, the heater was turned off, and the quartz boat was cooled to room temperature in an Ar atmosphere (1000 sccm) using a small fan. The samples were then removed, yielding graphene/diamond substrates coated with $Ga_2O_3$ films. The gallium precursor used was gallium acetylacetonate ($C_{15}H_{21}O_6Ga$) with a purity >99.99%. The required mass of $C_{15}H_{21}O_6Ga$ was calculated based on the desired gallium concentration and dissolved in deionized water and concentrated hydrochloric acid to prepare the precursor solution. The concentration of this precursor solution was set between 0.02 mol/L and 0.08 mol/L, while the HCl concentration was adjusted between 1% and 10%. The concentrated hydrochloric acid was added to create acidic conditions to assist precursor dissolution in deionized water—otherwise the precursor solution cannot react completely.

## Characterization techniques

The atomic-level structure of the films and their cross-sectional states were examined using TEM (FEI Talos F200S). The crystallographic orientation and alignment of the $Ga_2O_3$ films were determined through XRD (Rigaku Ultima IV), with a scanning angle (2θ) range of 10° to 70° and a step size of 0.02°. The surface morphology of the films was observed using SEM (Thermo Scientific Apreo 2C) combined with an energy dispersive spectrometer to analyze the elemental composition and distribution within the samples. XPS (Thermo Fisher Escalab Xi + ) was employed for both qualitative and quantitative analyses of surface elements, characterizing the elemental content on the film surface. AFM (Bruker Dimension R Icon TM) was conducted to characterize the surface topography of the samples in tapping mode, providing 2D and 3D morphological information and average roughness data. The interfacial coupling state between the material and substrate was characterized using temperature-dependent in-situ Raman spectroscopy (Horiba LabRAM HR Evolution), facilitating a comprehensive analysis of the crystal quality of the films. Supplemented our study with time-domain thermoreflectance (TDTR, Pioneer-ONE) measurements of both the thermal conductivity and thermal boundary resistance. For these tests, we deposited 100 nm Al films on all three

samples via e-beam evaporation, with a pump beam spot radius of 12 μm (at $1/e^2$ intensity) and a probe beam spot radius of 4 μm (at $1/e^2$ intensity), using a 1 MHz modulation frequency at 25 °C. By fitting the measurement curves with theoretical models, we obtained the $Ga_2O_3$ thermal conductivity parameters and $Ga_2O_3$-diamond thermal boundary resistance values.

## Adsorption energy calculation

All calculations were performed using the Adsorption Locator module in Materials Studio. The simulations were based on a force-field approach, and the adsorption energies of oxygen atoms on diamond surfaces were calculated using a Monte Carlo method combined with Simulated Annealing to identify the most probable adsorption sites. The Universal force field was employed to describe interatomic interactions, including van der Waals and electrostatic terms, with a cutoff distance of 12.5 Å for interactions. To avoid periodic boundary artifacts, a vacuum layer of 30 Å was added perpendicular to the surface. The convergence tolerances for energy and force were set to $1.0 \times 10^{-4}$ kcal/mol and 0.005 kcal/mol/Å, respectively. The displacement tolerance was set to $5.0 \times 10^{-5}$ Å, and the maximum number of iterations was set to 500. The binding energies (Eb) of oxygen atoms on three different 2D insertion layers grown on diamond substrates oriented along [100], [110], and [111] were calculated. The value of Eb was estimated using Eb = Etotal − Eo − Eslab, where Etotal, Eo, and Eslab denote the total energy after adsorption, energy of the oxygen atom monomer, and energy of the substrate system, respectively.

## Data availability

All data supporting the findings of this study are included within the paper and its Supplementary Information. All raw data generated in this study can be obtained from the corresponding author J.N. upon request and may be used for research purposes. Requests for access will typically be addressed within ten days. Source data are provided with this paper.

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

## Acknowledgements

J.N. acknowledges support from the General Program of Natural Science Foundation of China (Grant No: 62274134) and the National Key Research and Development Program (Grant Nos: 2023YFB3609900 and 2021YFA0716400); J.C.Z. acknowledges support from the National Science Foundation for Distinguished Young Scholars (Grant No: 62525402) and the National Science Foundation for Innovative Research Groups (Grant No: 62421005); J.N. acknowledges support from Key R&D Project in Xi'an City (Grant No: 2023JH-ZCGJ-0013); Aerospace Institute 771 Innovation Fund (Grant No:771CX2023007); The Natural Science Basic Research Program of Shaanxi Province (Grant No. 2025SYS-SYSZD-084).

## Author contributions

Z.C.Y. carried out the experiment, including grew the material and analyzed physicochemical characterization results. Z.C.Y., J.N. and D.W. designed the figures. Z.C.Y., H.D.W. and J.N. wrote the manuscript. Y.F.C., X.M.D., X.B.Z. and Y.N.Z. were involved in a discussion about device preparation and growth mechanism. J.C.Z. and Y.H. supervised the complete work. All authors discussed the results and remarked on the manuscript at all stages.

## Competing interests

The authors declare no competing interests.
