## [Transparent Peer Review file · Nature Communications]

Van der Waals β -Ga₂O₃ Thin Films on Polycrystalline Diamond Substrates

Corresponding Author: Professor Jing Ning

Version 0:

Reviewer comments:

Reviewer #1

(Remarks to the Author)

The authors present a study on integrating gallium oxide films with polycrystalline diamond using a ML graphene interlayer to reduce the stress coupling between the two materials. The authors motivate the study through the need for thermal management of wide bandgap semiconductors such as gallium oxide by their integration with substrates such as diamond. However, there are no good ways to grow gallium oxide on single-crystal diamond at scale. The method presented by the authors allows for integration with polycrystalline diamond at scale through the use of the ML graphene interlayer along with a unique growth method. This is the novelty of this work. However, there are several areas that have not been addressed to a sufficient level by the authors that the reviewer raises below:

- 1) The manuscript lacks a description of the layer thicknesses for the films that are presented in the manuscript. This helps to assess the integration scheme with the current body of knowledge that exists for gallium oxide electronics.
- 2) The authors mention that the thermal conductivity of the polycrystalline diamond was on the order of 95 W/mK, which is relatively low. Using microwave plasma CVD, films as thin as 1 micron in thickness can achieve this level of thermal conductivity. Also, the rms surface roughness is missing for the diamond, which impacts the integration and thermal transport properties of the interface. Please explain in more detail the source of the diamond, how it was grown, the rms surface roughness for the integration, and how it was measured.
- 3) The authors mention that the thermal conductivity of the gallium oxide was $10^{\wedge}30 \text{ W}\cdot\text{m}^{-1}\cdot\text{K}^{-1}$. Was this correct? Please provide a reference. (Did you mean this should be "between 10 - 30 $\text{W}\cdot\text{m}^{-1}\cdot\text{K}^{-1}$?)
- 4) The authors provide no measurement of the impact of the ML graphene on the thermal transport across the interface. This should include a measurement such as thermal boundary resistance with and without the ML graphene (or compare to other values in the literature for transport without graphene interlayers). At present, there is no way to tell if the ML graphene helps thermally or simply with the integration of gallium oxide on diamond. Again, there are no results (experiments or modeling) that show how much impact this integration scheme has over others being pursued in the field.
- 5) The authors show the Raman peak shift vs temperature in figures 5a and 5b. However, the authors fail to provide the shift coefficient and whether or not it's linear. The data in 5d are not symmetric as well. Is there an explanation for this? Please provide more data and a deeper explanation.
- 6) In Figure 5e, it is unclear why a localized hotspot of the shape and size depicted by the authors occurs. In addition, there is no link between this phenomenon, the diamond integration, and device thermal performance after Figure 5e. This leaves the manuscript somewhat lacking in its impact.
- 7) The hydroxyl groups on the graphene surface can impact the integration of gallium oxide with ML graphene. How was this controlled during the growth of samples?

Reviewer #2

(Remarks to the Author)

The manuscript under review is a very interesting and comprehensive study, combining experiment and theory, on a very hot topic today. It presents new original results and is well-organized, well-written and illustrated. The methodology is sound, and the manuscript meets the expected standards in the field. The work supports the conclusions and claims, there are no any flaws in the data analysis, interpretation and conclusions. Enough detail is provided in most methods for the work to be reproduced. The manuscript deserves publication in your journal since it of significance to the field and of high relevance to the journal audience. However, some points should be improved/better explained before acceptance according to the issues below:

1. Add the thickness of the thin films in the Abstract and Conclusions when commenting on the RMS and the FWHM of the "Rocking curve" values. Add "the smallest Rocking curve FWHM value of...".
2. Rewrite the first sentences of the introduction part starting with the advantageous properties and after that the device advancements because the term "emerging semiconductor" is not consistent with "advanced device fabrication". Add "radiation" stability [<https://doi.org/10.1016/j.vacuum.2022.111005>, etc.] to the thermal [Ref.] and chemical stability [Ref.] of beta-Ga₂O₃ if you do not comment on the other polytypes [DOI: 10.1109/JSEN.2023.3297127; DOI: 10.1063/5.0090832]. Better describe the advantages of beta-Ga₂O₃ compared to the classical and wide bandgap semiconductors for high-power electronics. Please include a few more references. Replace Ref. 1 with some for MOCVD beta-Ga₂O₃ [<https://doi.org/10.1063/1.5058059>; <https://doi.org/10.1116/6.0003424>] because the inorganic CVD method - HVPE - does not provide as high crystalline quality as the MOCVD. Ref. 3 and 5 are not for epitaxial thin films. Divide the optoelectronics [DOI: 10.1109/JSEN.2023.3297127; DOI: 10.1063/5.0090832] from high-power-electronics-related references.
4. If possible, shorten the figure captions.
5. You could omit the oxygen flow-rate in the Conclusions and slightly shorten the Conclusions including the most important achievements only.
6. In Fig. 3a there is a peak at about 63-degree theta. Can you make an assignment of this peak?
7. It is not mentioned in the Experimental part that the sample's surface is cleaned in the XPS chamber by ion sputtering. Does it mean the surface of the beta-Ga₂O₂ studied by XPS is as-grown? Do you expect some additional oxidation of the surface if the samples are stored in ambient conditions?
8. In Fig. 4c the bar is missing.
9. Please describe in detail the growth parameters of the employed mist-CVD approach (e.g., precursor concentration, HCl role, droplet size). Clarify how the precursor delivery and reaction kinetics are optimized or give some references.
10. The wet transfer method of Graphene onto polycrystalline diamond lacks critical details (e.g., etching solution, adhesion quality, defect density). Include some AFM/SEM images of the graphene/diamond interface.
11. Provide computational details (e.g., pseudopotentials, convergence criteria) to ensure reproducibility.
12. Shortly discuss scalability challenges (e.g., wafer-scale graphene uniformity, polycrystalline diamond grain boundaries) in terms of industrial application.
13. Comment on the long-term stability (e.g., interfacial degradation under high-power/frequency operation).
14. Is the dark current value of 5.83×10^{-6} μ A not rather low?

Reviewer #3

(Remarks to the Author)

Reviewer #4

(Remarks to the Author)

This manuscript presents an excellent study that demonstrates the interface stress relaxation, high crystallinity, enhanced thermal conductivity, and high-performance UV photodetection characteristics of β -Ga₂O₃ thin films. In particular, the authors clearly recognize the limitations of wafer-scale single-crystal diamond substrates and the structural defects associated with conventional heteroepitaxial approaches. Their adoption of the Van der Waals epitaxy (VdW epitaxy) strategy to overcome these issues is both innovative and practically meaningful.

If the following points can be further addressed or clarified, it will significantly enhance the completeness and scientific contribution of the paper. I recommend these aspects be considered before deciding on acceptance.

1. What are the specific advantages of mist-CVD-based VdW epitaxy compared to MOCVD, HVPE, or bonding methods? A comparative explanation, particularly regarding nucleation on chemically inert surfaces like graphene, would be helpful.

2. In the DFT calculations, does the use of a single-crystal vs. polycrystalline substrate significantly affect the calculated binding energy? Ensuring consistency with the actual experimental conditions is important for reliable interpretation.
3. Could the uniformity of the graphene transfer process—including potential defects, contamination, or mechanical strain—have influenced the crystallinity and device performance of the resulting Ga₂O₃ films? In particular, please discuss the correlation with grain boundary formation, such as those observed at 760 °C in Fig. 4(d).
4. Is the reported effective thermal conductivity value (95.8 W/m-K) derived with consideration of measurement uncertainty, contact resistance, and boundary conditions? A more detailed explanation would improve the reliability of the thermal analysis.
5. How might the variation in film thickness affect crystallinity and the measured effective thermal conductivity? Especially for thin films, could the high thermal conductivity of the underlying substrate lead to an overestimation of the intrinsic thermal properties of β-Ga₂O₃?

Version 1:

Reviewer comments:

Reviewer #1

(Remarks to the Author)

The authors have satisfied the concerns I raised in the first review.

Reviewer #2

(Remarks to the Author)

The authors have professionally answered the questions raised by all reviewers. The manuscript has been improved and in the present shape is acceptable for publication.

Reviewer #4

(Remarks to the Author)

This manuscript presents a well-structured and comprehensive study on the van der Waals epitaxy of β-Ga₂O₃ on polycrystalline diamond, demonstrating both scientific novelty and technical rigor. The authors have thoroughly addressed all reviewer comments with substantial experimental additions and clear revisions.

Response Letter

We sincerely thank the referees for their careful review and valuable comments, which certainly helped improve our manuscript titled “Van der Waals β -Ga₂O₃ Thin Films on Polycrystalline Diamond Substrates”. We also hope the editor could give us the opportunity, to reconsider the revised manuscript. Our point-by-point responses are presented below and all the changes in the revised manuscript have been highlighted in yellow for your review.

Response to the Reviewer 1

Overall Assessment from Reviewer 1:

The authors present a study on integrating gallium oxide films with polycrystalline diamond using a ML graphene interlayer to reduce the stress coupling between the two materials. The authors motivate the study through the need for thermal management of wide bandgap semiconductors such as gallium oxide by their integration with substrates such as diamond. However, there are no good ways to grow gallium oxide on single-crystal diamond at scale. The method presented by the authors allows for integration with polycrystalline diamond at scale through the use of the ML graphene interlayer along with a unique growth method. This is the novelty of this work. However, there are several areas that have not been addressed to a sufficient level by the authors that the reviewer raises below:

Response: We greatly appreciate the approval of our work. In the revised version, new comprehensive experimental data are included (describe in detail below) to address all concerns of the reviewer. In addition, we have also corrected known expression errors and grammatical errors.

Comment #1.1 The manuscript lacks a description of the layer thicknesses for the films that are presented in the manuscript. This helps to assess the integration scheme with the current body of knowledge that exists for gallium oxide electronics.

Response: We sincerely appreciate the valuable suggestion and apologize for the lack of description regarding the thickness of Ga₂O₃ thin films in the manuscript. Here, we

will discuss more details and all the changes in the revised manuscript have been highlighted in yellow for your review.

In this experiment, the Ga₂O₃ thin films were deposited via mist chemical vapor deposition (mist-CVD). A 35 mL precursor solution was used, and the deposition was carried out for 70 minutes at a rate of approximately 5 nm/min. As a result, the average thickness of the deposited films was ~350 nm. To confirm the film thickness, cross-sectional transmission electron microscopy (TEM) was performed, as shown in **Figure R1**. The TEM image clearly demonstrates that the film exhibits a uniform thickness of ~350 nm, consistent with the expected deposition parameters.

Figure R1. **a**, High-magnification cross-sectional TEM image of the interface between β -Ga₂O₃, graphene, and poly-diamond. **b**, High-magnification cross-sectional TEM image of the interface between VdW- β -Ga₂O₃, ML graphene, and polycrystalline diamond.

In the revised manuscript (Abstract, the sixth sentence), the authors added: “The 350 nm thick, high-crystallinity films exhibited a smallest Rocking curve FWHM value of 0.18° and a root mean square roughness of 6.71 nm.”

In the revised manuscript (Paragraph Six, Supplementary Fig.1), the authors added: “...where the 350 nm thick films (Supplementary Fig.1) exhibit both the highest crystallinity and fewest interfacial defects.”

In the revised manuscript (Conclusion, the eighth sentence.), the authors added: “...an RMS roughness of 6.71 nm at 350 nm thickness.”

Comment #1.2 The authors mention that the thermal conductivity of the polycrystalline diamond was on the order of 95 W/mK, which is relatively low. Using microwave plasma CVD, films as thin as 1 micron in thickness can achieve this level of thermal conductivity. Also, the rms surface roughness is missing for the diamond, which impacts the integration and thermal transport properties of the interface. Please explain in more detail the source of the diamond, how it was grown, the rms surface roughness for the integration, and how it was measured.

Response: We sincerely appreciate the reviewer's valuable suggestions regarding the need for more detailed information about the polycrystalline diamond substrate. Our research group possesses extensive expertise in diamond growth, having previously developed high-conductivity diamond via MPCVD (*IEEE Electron Device Lett.* **2023**, 45, 48-51)¹ and demonstrated high-performance radiation detectors (*Sci. China Mater.*, **2024**, 67, 2329-2334).² In this study, we employed Element Six's TM180 series polycrystalline diamond. The originally reported value of 95 W/m·K represents the off-chip thermal conductivity of Ga₂O₃ epitaxial layers on polycrystalline diamond, which we have now corrected and supplemented with additional data in the revised manuscript.

The technical specifications of the polycrystalline diamond substrate are sourced from Element Six *Diamond Handbook 2024*.³ The handbook details that the diamond growth method is microwave plasma chemical vapor deposition (MPCVD), where growth conditions are achieved through thermal dissociation of hydrogen and carbon-containing gas precursors in a high-temperature plasma (exceeding 2000°C). A schematic of the MPCVD system and growth mechanism are provided in **Figure R2**. The handbook specifies that its thermal conductivity exceeds 1800 W/m·K at 300 K and remains above 1500 W/m·K at 425 K. For polycrystalline CVD diamond, the surface roughness after polishing (Ra) is less than 20 nm (Element Six, *Diamond Handbook 2024*).

[Figure Redacted]

Figure R2. a, The schematic diagram of the diamond growth apparatus.³ **b**, Schematic of the diamond epitaxy with a compressed plasma ball. The microwave plasma with/without compression is shown using the solid/dashed lines.²

In the revised manuscript (Methods, first paragraph), the authors added: “We utilized polycrystalline diamond substrates from Element Six's TM180 series. According to the manufacturer's specifications (Element Six, *Diamond Handbook 2024*), these substrates demonstrate: Thermal conductivity >1800 W/m·K at 300 K, Thermal conductivity >1500 W/m·K at 425 K, Polished surface roughness (Ra) <20 nm. (Extended Data Supplementary Table 1)”

References:

1. Zhang, J. F., Zhang J.C., Hao, Y. et al. High Conductivity Hydrogenated Boron and Silicon Co-Doped Diamond With 0.46 Ω·mm Ohmic Contact Resistance. *IEEE Electron Device Letters*, **45**, 48-51. (2023).
2. Ding, S., Zhang, J. F., Zhang J.C. et al. Single-crystal diamond grown through high-power-density epitaxy used for a high-performance radiation detector. *Science China Materials*, **67**, 2329-2334. (2024).
3. Element Six, *CVD diamond handbook*, 2024. <https://e6cvd.com/us/diamond-book-download>.

Comment #1.3 The authors mention that the thermal conductivity of the gallium oxide was 10⁻³⁰ W·m⁻¹·K⁻¹. Was this correct? Please provide a reference. (Did you mean this should be "between 10 - 30 W·m⁻¹·K⁻¹?)

Response: We sincerely appreciate your careful review and apologize for the

inadvertent error in the thermal conductivity values. We have revised from “ 10^{-30} W/m·K” to “10-30 W/m·K”. (Highlighted in the revised manuscript).

In the revised manuscript (The second paragraph, the second sentence), the authors added: “One critical limitation is its relatively low thermal conductivity, approximately 10–30 W·m⁻¹·K⁻¹, which is only one-sixth that of diamond.”

Comment #1.4 The authors provide no measurement of the impact of the ML graphene on the thermal transport across the interface. This should include a measurement such as thermal boundary resistance with and without the ML graphene (or compare to other values in the literature for transport without graphene interlayers). At present, there is no way to tell if the ML graphene helps thermally or simply with the integration of gallium oxide on diamond. Again, there are no results (experiments or modeling) that show how much impact this integration scheme has over others being pursued in the field.

Response: We sincerely appreciate the reviewer’s insightful comments regarding heat transfer effects and apologize for our oversight. Over the past two decades, ultrafast laser-based time-domain thermoreflectance (TDTR) has emerged and evolved into a reliable, robust, and versatile technique (*Nat.* **2021**, 597, 660–665)⁴ for measuring the thermal properties of various thin-film materials and their interfaces (*Nat. Commun.* **2022**, 13, 7201).⁵ Here, we employed TDTR to measure the thermal conductivity of β -Ga₂O₃ thin films and the effective thermal boundary resistance (TBR_{eff}) at the Ga₂O₃ / diamond interface for three distinct samples: Sample 1 with a graphene / polycrystalline diamond (Gra/Dia) substrate, Sample 2 with a polycrystalline diamond (Dia) substrate, and Sample 3 with an O₂ plasma-treated graphene / polycrystalline diamond (P-Gra/Dia) substrate (50 W, 30 s). All samples were epitaxially grown under identical conditions to deposit β -Ga₂O₃ films.

The results, presented in **Figure R3a**, demonstrate that the directly epitaxial sample (Dia) exhibits the lowest TBR_{eff} at 2.82 m²K/GW, while Samples 1 and 3 show slightly higher values. This discrepancy arises from the van der Waals (vdW) interface, where thermal transport is limited by reduced effective contact area and weak adhesion

energy, leading to suppressed phonon transmission. In Sample 3, the plasma-treated graphene introduces a small fraction of covalent bonds at the interface, marginally reducing thermal resistance. However, the plasma also causes random nucleation, degrading the crystalline quality of the Ga₂O₃ film and resulting in a slightly lower thermal conductivity. Consequently, the overall trend follows TC(P-Gra/Dia) < TC(Gra/Dia) < TC(Dia).

A comparison with contemporary international benchmarks (**Figure R3b**) reveals that our approach achieves the lowest reported TBR_{eff}—an order of magnitude lower than previous bonding or interlayer strategies. This advancement builds upon our group’s extensive expertise in vdW epitaxy, as evidenced by prior successes in high-quality AlN (*CrystEngComm*, **2021**, 23, 7406–7411)⁶ and GaN (*Cryst. Growth Des.*, **2021**, 21, 5848–5853)⁷ epitaxial films, alongside applications in deep-UV LEDs (*Nanotechnology*, **2023**, 34, 295202)⁸ and solid-state lighting (*ACS Appl. Nano Mater.*, **2020**, 3, 5061–5069).⁹ Leveraging this robust foundation, our current work addresses the critical heat dissipation requirements for kW-scale power devices.

Figure R3. **a**, Thermal conductivity of Ga₂O₃ films and interfacial thermal boundary resistance (TBR_{eff}) for sample1, 2 and 3. **b**, Benchmark comparison of TBR_{eff} values across different interface engineering approaches.

[1] Xu, W. et al. Thermal transport properties of β-Ga₂O₃ thin films on Si and SiC substrates fabricated by an ion-cutting process. *ACS Applied Electronic Materials*, **6**, 1710-1717 (2024).

[2] Zhao, T. et al. First Demonstration of Wafer-Level Arrayed β-Ga₂O₃ Thin Films and

MOSFETs on Diamond by Transfer Printing Technology. *IEEE International Electron Devices Meeting*, pp. 1-4 (2024).

[3] Malakoutian, M. et al. Polycrystalline diamond growth on β -Ga₂O₃ for thermal management. *Applied Physics Express*, **14**, 055502 (2021).

[4] Nepal, N. et al. Heteroepitaxial growth of β -Ga₂O₃ films on SiC via molecular beam epitaxy. *Journal of Vacuum Science & Technology A*, **38**, 063406. (2020).

[5] Cheng, Z., Yates, L., Shi, J., Tadjer, M. J., Hobart, K. D., & Graham, S. Thermal conductance across β -Ga₂O₃-diamond van der Waals heterogeneous interfaces. *Appl Materials*, **7**, 031118. (2019).

[6] Cheng, Z. et al. Thermal transport across ion-cut monocrystalline β -Ga₂O₃ thin films and bonded β -Ga₂O₃-SiC interfaces. *ACS Applied Materials & Interfaces*, **12**, 44943-44951. (2020).

[7] Song, Y. et al. Ga₂O₃-on-SiC composite wafer for thermal management of ultrawide bandgap electronics. *ACS Applied Materials & Interfaces*, **13**, 40817-40829. (2021).

[8] Vaca, D. et al. Thermal conductivity of β -Ga₂O₃ thin films grown by molecular beam epitaxy. *ITherm*, pp. 1011-1016. (2020).

In the revised manuscript (Paragraph Eleven), the authors added: “Here, we systematically measured the thermal conductivity of Ga₂O₃ films and the effective thermal boundary resistance (TBR_{eff}) at Ga₂O₃/Diamond interfaces using time-domain thermoreflectance (TDTR) for three samples: Sample 1(Graphene/poly-diamond, Gra/Dia), Sample 2(bare poly-diamond, Dia), and Sample 3(O₂ plasma-treated Graphene/poly-diamond, P-Gra/Dia, 50W for 30s). All samples underwent identical β -Ga₂O₃ epitaxy conditions. As shown in Figs. 5c, Sample 2 exhibited the lowest effective thermal boundary resistance of 2.82 m²K/GW, while Sample 1 and Sample 3 showed higher values of 6.08 m²K/GW and 6.02 m²K/GW respectively. This discrepancy arises from the limited phonon transport across vdW interfaces, governed by reduced actual contact area and weak adhesion energy. Although the plasma-treated graphene introduced sparse covalent bonds (marginally lowering TBR_{eff}), the random nucleation induced by nonuniform plasma treatment degraded film crystallinity, resulting in slightly lower Ga₂O₃ thermal conductivity. Consequently, the overall trend followed:

$TC(\text{P-Gra/Dia}) < TC(\text{Gra/Dia}) < TC(\text{Dia})$. As demonstrated by Zheng et al., the thermal conductivity of Ga_2O_3 with specific orientations increases with thickness.¹¹ In our experiments, the $(\bar{2}01)$ plane rocking curve showed a minimum FWHM of 0.18° for the 350-nm-thick film, corresponding to a thermal conductivity of $7.19 \text{ W/m}\cdot\text{K}$. Furthermore, thinner films display more pronounced phonon boundary scattering (consistent with the Callaway model), leading to reduced effective thermal conductivity. As presented in Fig. 5d, our results demonstrate a TBR_{eff} that is one order of magnitude lower than previous bonding and interlayer approaches, which is attributed to the ultraclean interface processing and optimized graphene transfer enabling atomically flat interfaces. This breakthrough fulfills the thermal management requirements for kW-class power devices.”

In the revised manuscript (Abstract), the authors added: “ $\beta\text{-Ga}_2\text{O}_3/\text{diamond}$ interface exhibits an ultralow thermal boundary resistance of $2.82 \text{ m}^2\cdot\text{K}/\text{GW}$.”

In the revised manuscript (Conclusions, the fifth sentence from the end), the authors added: “TDTR measurements reveal an ultralow TBR_{eff} of $2.82 \text{ m}^2\text{K}/\text{GW}$ at the $\beta\text{-Ga}_2\text{O}_3/\text{diamond}$ interface, representing a one-order-of-magnitude reduction compared with state-of-the-art reported values.”

References:

4. Kim, S. E. et al. Extremely anisotropic van der Waals thermal conductors. *Nature*, **597**, 660-665. (2021).
5. Cheng, Z. et al. High thermal conductivity in wafer-scale cubic silicon carbide crystals. *Nature communications*, **13**, 7201. (2022).
6. Jia Y, Wu H, Zhao J, et al. Growth mechanism on graphene-regulated high-quality epitaxy of flexible AlN film. *CrystEngComm*, **23**, 7406-7411. (2021).
7. Wu H, Ning J, Jia Y, et al. Van der waals self-assembled silica-nanosphere/graphene buffer layer for high-quality gallium nitride growth. *Crystal Growth & Design*, **21**, 5848-5853. (2021)
8. Wu H, Ning J, Zhang J, et al. High quality AlN film assisted by graphene/sputtered AlN buffer layer for deep-ultraviolet-LED. *Nanotechnology*, **34**, 295202. (2023).
9. Ning J, Yan C, Jia Y, et al. GaN films deposited on sapphire substrates sputter-coated

with AlN followed by monolayer graphene for solid-state lighting. *ACS Applied Nano Materials*, **3**, 5061-5069. (2020).

Comment #1.5 The authors show the Raman peak shift vs temperature in figures 5a and 5b. However, the authors fail to provide the shift coefficient and whether or not it's linear. The data in 5d are not symmetric as well. Is there an explanation for this? Please provide more data and a deeper explanation.

Response: We sincerely appreciate the reviewer's thorough examination of our Raman data. The temperature-dependent Raman peak shifts were analyzed to demonstrate how the graphene interlayer mitigates in-plane thermal stress between diamond and Ga₂O₃ caused by their significant thermal expansion coefficient mismatch. The weak van der Waals bonding at the graphene interface effectively reduces thermal stress-induced damage to the Ga₂O₃ film at elevated temperatures. Our primary objective was to compare the Raman peak red shifts ($\Delta\omega$) of β -Ga₂O₃ on Si, sapphire, and diamond substrates during thermal cycling to evaluate the tensile stress in the films. (*Nat.* **2020**, 577, 204-208)¹⁰ Therefore, the trend of $\Delta\omega$ with temperature and its linearity are not particularly relevant to this specific evaluation.

Here, we present a comparison of Raman (**FigureR4a and 4b**) shift magnitudes at different temperatures, as shown in **FigureR4c and 4d**, the largest $\Delta\omega$ occurred for β -Ga₂O₃ on diamond at 100°C, indicating the highest in-plane tensile stress, which was slightly alleviated by the graphene interlayer. The smaller $\Delta\omega$ difference between sapphire and Si substrates ($\Delta\omega_{\text{Si},100^\circ\text{C}} = 2.418 \text{ cm}^{-1}$ vs $\Delta\omega_{\text{Sap},100^\circ\text{C}} = 2.413 \text{ cm}^{-1}$) reflects their closer thermal expansion coefficients to β -Ga₂O₃. When the temperature was increased to 200°C, the $\Delta\omega$ of β -Ga₂O₃ on polycrystalline diamond continued to rise to 10.8 cm⁻¹, yielding a calculated temperature coefficient of peak shift of approximately -0.05 cm⁻¹/K over the heating range. In contrast, β -Ga₂O₃ on both Si and sapphire substrates exhibited a smaller temperature coefficient of -0.024 cm⁻¹/K, which can be attributed to their relatively smaller thermal expansion coefficient mismatches.

Figure R4. **a**, Typical Raman spectra of $\beta\text{-Ga}_2\text{O}_3$ epitaxially grown on ML-graphene/polycrystalline diamond and **b**, local magnified Raman spectra. **c**, comparative data of Raman peak red shifts ($\Delta\omega$) in $\beta\text{-Ga}_2\text{O}_3$ on different substrates at 100°C and **d**, 200°C, respectively.

Additionally, the brief temperature holding periods during heating and cooling cycles prevented complete recovery of the interfacial state to its initial condition, resulting in the asymmetric data in Figure 5d as noted by the reviewer. Importantly, the complete recovery of Raman spectral shifts demonstrates that the thermal cycling process is non-destructive to $\beta\text{-Ga}_2\text{O}_3$. The slight blue shift observed at the final 0°C measurement for $\beta\text{-Ga}_2\text{O}_3$ on Gra/Dia, Si, and Sapphire substrates is likely attributable to enhanced interfacial bonding strength induced by high-temperature treatment, which generates localized compressive stress and consequently increases phonon vibrational frequencies.

In the revised manuscript (Paragraph 10, Sentence 8), the authors added: “The smaller $\Delta\omega$ difference between sapphire and Si substrates ($\Delta\omega_{\text{Si},100^\circ\text{C}} = 2.418 \text{ cm}^{-1}$ vs $\Delta\omega_{\text{Sap},100^\circ\text{C}} = 2.413 \text{ cm}^{-1}$) reflects their closer thermal expansion coefficients to $\beta\text{-Ga}_2\text{O}_3$.”

Ga₂O₃.(Supplementary Fig.5) When the temperature was increased to 200°C, the $\Delta\omega$ of β -Ga₂O₃ on polycrystalline diamond continued to rise to 10.8 cm⁻¹, yielding a calculated temperature coefficient of peak shift of approximately -0.05 cm⁻¹/K over the heating range. In contrast, β -Ga₂O₃ on both Si and sapphire substrates exhibited a smaller temperature coefficient of -0.024 cm⁻¹/K, which can be attributed to their relatively smaller thermal expansion coefficient mismatches. The slight blue shift observed at the final 0°C measurement for β -Ga₂O₃ on Gra/Dia, Si, and Sapphire substrates is likely attributable to enhanced interfacial bonding strength induced by high-temperature treatment, which generates localized compressive stress and consequently increases phonon vibrational frequencies.”

References:

10. Yuan, G. et al. Proton-assisted growth of ultra-flat graphene films. *Nature*, **577**, 204-208 (2020).

Comment #1.6 In Figure 5e, it is unclear why a localized hotspot of the shape and size depicted by the authors occurs. In addition, there is no link between this phenomenon, the diamond integration, and device thermal performance after Figure 5e. This leaves the manuscript somewhat lacking in its impact.

Response: We sincerely appreciate the reviewer's valuable suggestions for improving our manuscript. Regarding Figure 5e, we originally intended to illustrate both the structure of our fabricated photodetector and its thermal management characteristics. However, as rightly pointed out by the reviewer, this schematic lacked sufficient scientific rigor and coherence with the overall manuscript. In response, we have removed the localized hotspot illustration from the revised manuscript to improve the logical flow and consistency of the paper.

Comment #1.7 The hydroxyl groups on the graphene surface can impact the integration of gallium oxide with ML graphene. How was this controlled during the growth of samples?

Response: We sincerely appreciate the reviewer's insightful questions. Our research

group has conducted extensive studies on graphene and achieved significant advancements in vdW epitaxy, enabling stress-free nitride thin film growth (*Appl. Sci.*, **2020**, 10, 8814).¹¹ These efforts have led to the development of high-brightness violet light-emitting diodes (*Adv. Opt. Mater.*, **2020**, 8, 1901632)¹² and high-performance light-emitting diodes (*ACS Appl. Mater. Interfaces*, **2021**, 13, 32442–32449)¹³. Our foundational work in vdW heteroepitaxy provides critical insights for optimizing thermal and electronic properties in wide-bandgap semiconductor devices.

Therefore, we have previously investigated methods to control the influence of surface hydroxyl groups on epitaxial materials during graphene transfer. Methyl methacrylate (MMA) was employed as the transfer support layer due to its exceptionally low adsorption energy with graphene and minimal atomic hybridization, which significantly reduces residue contamination (*Carbon*, **2020**, 169, 92-98).¹⁴ Compared to conventional PMMA transfer, the MMA-assisted process yields graphene with lower surface roughness and fewer residues, as confirmed by SEM and AFM analysis, thereby substantially reducing the likelihood of surface contaminants such as hydroxyl groups. As demonstrated in **Figure R5a**, Raman spectroscopy reveals no detectable D peak (defect-related) in the transferred graphene, indicating negligible defects and impurities including hydroxyl groups. XPS analysis further verifies the absence of MMA residues and confirms the chemically pristine state of the surface.

For the current experiment, the copper foil was pre-treated by electrochemical polishing prior to CVD growth. After growth, the copper substrate was etched using an ammonium persulfate ((NH₄)₂S₂O₈) solution, followed by multiple rinses with deionized water to remove residual etchant. Before transferring graphene onto the polycrystalline diamond substrate, the diamond was sequentially cleaned with acetone, ethanol, and deionized water via ultrasonication, followed by a 5-minute immersion in dilute HF solution to eliminate surface oxides, ensuring an atomically clean interface. (**Figure R5b**)

[Figure Redacted]

Figure R5. a, Displacements of G peaks and 2D peaks of the monolayer and bilayer graphene films, changes in FWHM of G peaks and 2D peaks of the monolayer and bilayer graphene films; XPS spectra of the MTG and PTG.¹⁴ **b,** SEM and AFM images of the polycrystalline diamond substrate and wet-transferred monolayer graphene on polycrystalline diamond.

In the revised manuscript (Methods, First Paragraph), the authors added: “Following CVD growth, we employed methyl methacrylate (MMA) as a transfer support layer. The copper substrate was etched using an ammonium persulfate ((NH₄)₂S₂O₈) solution, followed by multiple rinses with deionized water to completely remove residual etchant, thereby minimizing interfacial hydroxyl (-OH) group formation and its impact on material properties.⁴¹ Prior to transferring graphene onto the polycrystalline diamond substrate, we performed ultrasonic cleaning of the diamond substrate using acetone, alcohol, and deionized water sequentially. Additionally, the substrate was immersed in dilute HF solution for five minutes to eliminate surface oxides, ensuring an atomically clean interface.”

References:

11. Zeng Y, Ning J, Zhang J, et al. Raman analysis of E2 (high) and A1 (LO) phonon to the stress-free GaN grown on sputtered AlN/graphene buffer layer. *Applied Sciences*, **10**, 8814. (2020).
12. Jia Y, Ning J, Zhang J, et al. Transferable GaN enabled by selective nucleation of AlN on graphene for high-brightness violet light-emitting diodes. *Advanced Optical*

Materials, **8**, 1901632. (2020).

13. Jia Y, Ning J, Zhang J, et al. High-quality transferred GaN-based light-emitting diodes through oxygen-assisted plasma patterning of graphene. *ACS Applied Materials & Interfaces*, **13**, 32442-32449. (2021).

14. Shen, X. et al. MMA-enabled ultraclean graphene transfer for fast-response graphene / GaN ultraviolet photodetectors. *Carbon*, **169**, 92-98 (2020).

Response to the Reviewer 2

Overall Assessment from Reviewer 2:

The manuscript under review is a very interesting and comprehensive study, combining experiment and theory, on a very hot topic today. It presents new original results and is well-organized, well-written and illustrated. The methodology is sound, and the manuscript meets the expected standards in the field. The work supports the conclusions and claims, there are no any flaws in the data analysis, interpretation and conclusions. Enough detail is provided in most methods for the work to be reproduced. The manuscript deserves publication in your journal since it of significance to the field and of high relevance to the journal audience. However, some points should be improved/better explained before acceptance according to the issues below:

Response: We greatly appreciate the reviewer's recognition of our work. In the revised version, new comprehensive experimental data and related statements (see below for details) are included to address all concerns of the reviewers. In addition, we have corrected known expression and grammar errors and accepted all insightful comments from the reviewers.

Comment #2.1 Add the thickness of the thin films in the Abstract and Conclusions when commenting on the RMS and the FWHM of the “Rocking curve” values. Add “the smallest Rocking curve FWHM value of...”.

Response: We appreciate the reviewer's constructive comments and apologize for any inappropriate expressions in manuscript. We have carefully emphasized the film thickness data in both the abstract and conclusions sections, while also explicitly

incorporating thickness references when discussing RMS roughness and rocking curve measurements. Additionally, we have modified the original phrase "The minimum FWHM" to the more precise technical description "the smallest rocking curve FWHM value of..." throughout the revised manuscript.

In the revised manuscript (Abstract, the sixth sentence), the authors added: "The 350 nm thick, high-crystallinity films exhibited a smallest rocking curve FWHM value of 0.18° and a root mean square roughness of 6.71 nm."

In the revised manuscript (Paragraph Six), the authors added: "The smallest rocking curve FWHM value of 0.18° occurs at 760°C..."

In the revised manuscript (Conclusion, the eighth sentence.), the authors added: "... the smallest rocking curve FWHM of 0.18° ..."

Comment #2.2 Rewrite the first sentences of the introduction part starting with the advantageous properties and after that the device advancements because the term "emerging semiconductor" is not consistent with "advanced device fabrication". Add "radiation" stability [<https://doi.org/10.1016/j.vacuum.2022.111005>, etc.] to the thermal [Ref.] and chemical stability [Ref.] of beta-Ga₂O₃ if you do not comment on the other polytypes [DOI: 10.1109/JSEN.2023.3297127; DOI: 10.1063/5.0090832]. Better describe the advantages of beta-Ga₂O₃ compared to the classical and wide bandgap semiconductors for high-power electronics. Please include a few more references. Replace Ref. 1 with some for MOCVD beta-Ga₂O₃ [<https://doi.org/10.1063/1.5058059>; <https://doi.org/10.1116/6.0003424>] because the inorganic CVD method - HVPE - does not provide as high crystalline quality as the MOCVD. Ref. 3 and 5 are not for epitaxial thin films. Divide the optoelectronics [DOI: 10.1109/JSEN.2023.3297127; DOI: 10.1063/5.0090832] from high-power-electronics-related references.

Response: We sincerely appreciate the suggestion to enhance the discussion of recent Ga₂O₃ advancements. The references proposed by the reviewer has significant influence in the industry and has generated tremendous impetus for the development of related fields, providing academic reference for this manuscript. The following key

references have been incorporated into the introduction section with expanded analysis:

1. Zhang, Y., Alema, F., Mauze, A., Koksaldi, O. S., Miller, R., Osinsky, A., & Speck, J. S. MOCVD grown epitaxial β -Ga₂O₃ thin film with an electron mobility of 176 cm²/V s at room temperature. *APL Materials*, **7** (2019).
2. Gogova, D., et al. High crystalline quality homoepitaxial Si-doped β -Ga₂O₃ (010) layers with reduced structural anisotropy grown by hot-wall MOCVD. *Journal of Vacuum Science & Technology A*, **42** (2024).
3. Titov, A. I., Karabeshkin, K. V., Struchkov, A. I., Nikolaev, V. I., Azarov, A., Gogova, D. S., & Karaseov, P. A. Comparative study of radiation tolerance of GaN and Ga₂O₃ polymorphs. *Vacuum*, **200**, 111005 (2022).
4. Polyakov, A., et al. Electrical properties of α -Ga₂O₃ films grown by halide vapor phase epitaxy on sapphire with α -Cr₂O₃ buffers. *Journal of Applied Physics*, **131** (2022).
5. Almaev, A., et al. Solar-blind ultraviolet detectors based on high-quality HVPE α -Ga₂O₃ films with giant responsivity. *IEEE Sensors Journal*, **23**, 19245-19255 (2023)

In the revised manuscript (First Paragraph), the authors added: “As a novel ultra-wide bandgap semiconductor, gallium oxide (Ga₂O₃) has attracted extensive research interest due to its rich material properties^{1,2}, making it an ideal candidate for next-generation high-power electronic devices³ and ultraviolet optoelectronic applications⁴. The β -phase Ga₂O₃ exhibits a breakdown electric field 27 times higher than silicon and 2.4 times greater than GaN, demonstrating exceptional suitability for ultra-high voltage MOSFETs and Schottky diodes⁵. With a Baliga's figure of merit (BFOM) 3000 times superior to silicon and 4 times better than GaN⁶, β -Ga₂O₃-based devices achieve significantly lower conduction loss and higher efficiency at equivalent voltage ratings. Owing to its outstanding properties including high breakdown field strength, low energy loss, excellent radiation hardness⁷, thermal stability⁸, and chemical stability⁹, β -Ga₂O₃ outperforms conventional wide-bandgap semiconductors by orders of magnitude in power electronics applications.”

Comment #2.3 If possible, shorten the figure captions.

Response: We sincerely appreciate the reviewer's constructive suggestions regarding our figure captions. In response to these valuable comments, we have carefully streamlined all figure captions to enhance clarity and conciseness, with all modifications clearly highlighted in yellow throughout the revised manuscript. These refinements have effectively improved the overall readability of the paper by eliminating unnecessary redundancies.

In the revised manuscript (The first figure caption), the authors revised: “Fig. 1. Adsorption energy calculations of VdW- β -Ga₂O₃ epitaxy on polycrystalline diamond. a, Atomic structures of O atoms adsorbed on ML-graphene/diamond, ML-h-BN/diamond, diamond substrate visualized. The orientations of the diamond are [100], [110], and [111]. Both top and side views are provided. b, Adsorption energies of oxygen atoms for the configurations depicted in panel a. c, Schematics illustrating the atomic structures of epitaxial Ga₂O₃ bulk material with ML-h-BN and Graphene as insertion layers. d, Adsorption energy trends for epitaxial Ga₂O₃ and e, O atoms as a function of the number of graphene and h-BN layers, with the number of insertion layers ranging from 1 to 6.”

In the revised manuscript (The second figure caption), the authors revised: “Fig. 2. TEM images of VdW- β -Ga₂O₃ on poly-diamond substrates. a, High-magnification cross-sectional TEM image of the interface between VdW- β -Ga₂O₃, ML graphene, and poly-diamond. b, HR-TEM images of the ($\bar{2}01$) β -Ga₂O₃ and c, ($\bar{4}01$) β -Ga₂O₃ crystal orientations, with corresponding atomic structure schematics. d, Crystal interface between the ($\bar{2}01$) β -Ga₂O₃ and ($\bar{4}01$) β -Ga₂O₃ orientations, including the measured interplanar spacings. e, Top-view atomic structures of the ($\bar{2}01$) β -Ga₂O₃ orientation and (001) graphene.”

In the revised manuscript (The fifth figure caption), the authors revised: “Fig. 5. Thermal dissipation characterization of VdW- β -Ga₂O₃ on polycrystalline diamond. a, Typical Raman spectra of β -Ga₂O₃ epitaxially grown on ML-graphene/polycrystalline diamond. b, Local magnified Raman spectra. c, Temperature-dependent Raman shift ($\Delta\omega$) of VdW- β -Ga₂O₃ on different substrate materials, extracted from in-situ Raman

measurements. d, Schematic of the photodetector structure and heat dissipation pathways. e, Thermal conductivity of Ga₂O₃ films and interfacial thermal boundary resistance (TBR_{eff}) for sample 1, 2 and 3. f, Photocurrent and dark current of the photodetector under 0.795 mW/cm² illumination. (Inset: optical microscope image of interdigitated electrodes). g, Comparison of the PDCR and responsivity of this photodetector against the performance metrics of other devices during the same period. h, Benchmark comparison of TBR_{eff} values across different interface engineering approaches.”

Comment #2.4 You could omit the oxygen flow-rate in the Conclusions and slightly shorten the Conclusions including the most important achievements only.

Response: We appreciate the reviewer's constructive suggestions for improving our conclusions. Based on the key finding that sufficient oxygen flow determines the ($\bar{2}01$) crystal orientation due to its highest lattice oxygen surface density, we have revised our conclusion to the more precise statement. This modified formulation maintains all original technical content while significantly improving readability.

In the revised manuscript (Conclusion, the fourth sentence), the authors revised: “Tunable growth of ($\bar{2}01$) VdW- β -Ga₂O₃ was achieved by controlling the lattice mismatch between graphene and the oxygen densities of different crystal orientations.”

Comment #2.5 In Fig. 3a there is a peak at about 63-degree theta. Can you make an assignment of this peak?

Response: We sincerely appreciate the reviewer's careful examination of the XRD patterns. As shown in **Figure R6**, our re-examination against the standard XRD cards revealed that the peak at 62.18° corresponds to the (009) orientation of the α -phase, which occurs because the temperature of 700°C remains relatively low and insufficient to meet the conditions for irreversible transformation of the metastable α -phase into the β -phase at higher temperatures. This peak completely disappears when the temperature exceeds 700°C. Additionally, sharp, zero-width peaks marked in blue were observed under 740°C and 720°C conditions but were absent at other temperatures, suggesting

these "needle-like peaks" may represent instrumental noise or artifacts. We have included this supplementary explanation in the revised manuscript.

Figure R6. XRD spectra of VdW-β-Ga₂O₃ thin films grown at different temperatures

Comment #2.6 It is not mentioned in the Experimental part that the sample's surface is cleaned in the XPS chamber by ion sputtering. Does it mean the surface of the beta-Ga₂O₃ studied by XPS is as-grown? Do you expect some additional oxidation of the surface if the samples are stored in ambient conditions?

Response: We appreciate the reviewer's meticulous examination of our XPS data. As correctly noted, surface contaminants such as hydrocarbons and adsorbed oxygen may remain if proper cleaning isn't performed prior to XPS measurements, potentially obscuring the intrinsic chemical states of β-Ga₂O₃. In our original study, to prevent such artifacts, all post-growth samples were systematically stored in nitrogen cabinets, with vacuum-sealed packaging used during any brief transfers. Crucially, all XPS characterizations were conducted within 12 hours of growth under these controlled storage conditions.

To allay doubts, we performed new comparative XPS measurements between freshly prepared samples and those stored under ambient conditions for 48 hours

(**Figure R7**). Spectral deconvolution of the O1s peaks reveals that O_{Lat} corresponds to lattice oxygen in Ga₂O₃ while O_{Vac} relates to oxygen vacancies. Quantitative analysis shows the oxygen vacancy concentration O_{Vac} / (O_{Lat} + O_{Vac}) decreased marginally from 32.8% (fresh) to 30.6% (48-hour ambient storage)

In summary, while minor surface oxidation variations exist among the samples, they do not affect the oxygen vacancy trend reported in the original manuscript. Furthermore, all samples were stored under strictly oxygen-free conditions, effectively eliminating any potential influence from ambient oxygen exposure.

Figure R7. O1s XPS spectra of **a**, as-grown β -Ga₂O₃ and **b**, after 48-hour ambient air exposure.

In the revised manuscript (The sixth paragraph, the ninth sentence), the authors added: “Our rigorous sample storage protocol ensures the temporal validity of the acquired data. (Supplementary Fig. 2)”

Comment #2.7 In Fig. 4c the bar is missing.

Response: We sincerely appreciate the reviewer's careful examination of the AFM images and apologize for our oversight. As shown in **Figure R8**, we have now added the scale bar, with this modification clearly highlighted in yellow throughout the revised manuscript.

Figure R8. AFM surface morphology images at 740°C, 760°C, and 780°C, showing the dependence of sample roughness on deposition temperature.

Comment #2.8 Please describe in detail the growth parameters of the employed mist-CVD approach (e.g., precursor concentration, HCl role, droplet size). Clarify how the precursor delivery and reaction kinetics are optimized or give some references.

Response: We sincerely appreciate the reviewer's insightful questions regarding our mist-CVD experiments. In fact, our research group has extensive expertise in mist-CVD technology, as demonstrated by our earlier achievements in growing high-quality β -Ga₂O₃ films on various substrates including GaN (*IEEE Trans. Electron Devices*, **2022**, 69, 1196-1199)¹⁵, sapphire (*Mater. Today Commun.*, **2021**, 29, 102766)¹⁶, and NiO (*Sci. China Mater.*, **2024**, 67, 1646-1653)¹⁷. Furthermore, we successfully developed solar-blind ultraviolet photodetectors and other functional devices using this technique as early as 2018 (*Opt. Mater. Express*, **2018**, 8, 2941-2947).¹⁸

The experimental setup is illustrated in **Figure R9a**. In the mist chemical vapor deposition (mist-CVD) process, an ultrasonic nebulizer atomizes the gallium acetylacetonate precursor solution into fine droplets, which are then transported into the reaction chamber by a carrier gas (**Figure R9b**). The working principle relies on the nebulizer's ability to convert the solution into micron-sized droplets, which are carried into the heated reaction chamber by oxygen carrier gas. Due to the Leidenfrost effect, these droplets levitate and migrate toward the heated substrate surface. When contacting the high-temperature surface, the bottom portion of each droplet vaporizes

immediately, creating a vapor layer that prevents direct liquid-substrate contact. Under the combined action of gravity and this vapor cushion, the droplets hover above the surface until complete evaporation occurs, enabling gradual thin film deposition. This unique transport mechanism allows for both precursor delivery and controlled vaporization at the substrate interface.

[Figure Redacted]

Figure R9. a, Schematic illustration of the mist-CVD experimental setup.¹⁹ **b**, Working principle diagram of the mist-CVD process developed by our research group.²⁰

For this experiment, we used gallium acetylacetonate ($C_{15}H_{21}O_6Ga$) from Aladdin with purity >99.99% as the gallium precursor for Ga_2O_3 thin film growth. We calculated the required mass of gallium acetylacetonate based on the desired gallium concentration and prepared the precursor solution by adding deionized water and concentrated hydrochloric acid. The precursor solution concentration used in this experiment was 0.02 mol/L. The HCl content was 1%, and the concentrated hydrochloric acid was added to create acidic conditions to assist precursor dissolution in deionized water - otherwise the precursor solution cannot react completely. We used 1.7 MHz ultrasonic atomization with oxygen (O_2) as the carrier gas and argon (Ar) as the dilution gas to transport the small mist droplets onto graphene/poly-diamond. Here, the O_2 flow rate was 600 sccm and Ar was 3000 sccm. The optimal O_2 flow rate was selected based on actual experimental data of material oxygen vacancies and comprehensive RMS values.

In the revised manuscript (Methods, the second paragraph), the authors added: "The working principle utilizes the atomization characteristics of the ultrasonic nebulizer to convert the solution into small droplets, which are then transported into the heated reaction chamber by oxygen carrier gas. Due to the Leidenfrost effect, the droplets can float and migrate to the surface of the heated substrate. When droplets land on the high-temperature surface, the portion contacting the substrate vaporizes under

heat, forming a vapor layer between them. This layer prevents direct contact, and under the combined action of gravity and the vapor layer, the droplets remain suspended on the surface until complete vaporization, ultimately forming a thin film.”

References:

15. Xu, Y., Zhang J, et al. Depletion-Mode β -Ga₂O₃ MOSFETs Grown by Nonvacuum, Cost-Effective Mist-CVD Method on Fe-Doped GaN Substrates. *IEEE Transactions on Electron Devices*, **69**, 1196-1199. (2022).
16. Cheng, Y., Zhang J, et al. Heteroepitaxial growth of β -Ga₂O₃ thin films on c-plane sapphire substrates with β -(Al_xGa_{1-x})₂O₃ intermediate buffer layer by mist-CVD method. *Materials Today Communications*, **29**, 102766. (2021).
17. Zhang, Z., Zhang J, et al. High-quality crystalline NiO/ β -Ga₂O₃ p–n heterojunctions grown by the low-cost and vacuum-free mist-CVD for device applications. *Science China Materials*, **67**, 1646-1653. (2024).
18. Xu, Y., Zhang J, et al. Solar blind deep ultraviolet β -Ga₂O₃ photodetectors grown on sapphire by the Mist-CVD method. *Optical Materials Express*, **8**, 2941-2947. (2018).
19. Zhu, X., et al. Atmospheric and aqueous deposition of polycrystalline metal oxides using Mist-CVD for highly efficient inverted polymer solar cells. *Nano letters*, **15**, 4948-4954. (2015).
20. Cheng, Y., Zhang J, et al. Heteroepitaxial growth of α -Ga₂O₃ thin films on a-, c-and r-plane sapphire substrates by low-cost mist-CVD method. *Journal of Alloys and Compounds*, **831**, 154776. (2020).

Comment #2.9 The wet transfer method of Graphene onto polycrystalline diamond lacks critical details (e.g., etching solution, adhesion quality, defect density). Include some AFM/SEM images of the graphene/diamond interface.

Response: We sincerely appreciate the reviewers’ constructive suggestions regarding the wet-transfer experiment section, which have now been incorporated into the revised manuscript. Our group has extensive expertise in epitaxial growth of high-quality thin films using 2D materials as intercalation layers, as demonstrated in our prior work

(*Small* **2021**, 17, 2105207).²¹ This experience has established a robust foundation for graphene wet-transfer techniques (*Sci. China Mater.* **2023**, 66, 1968–1977).²² Leveraging this knowledge, we optimized the transfer process by employing MMA as a support layer and fine-tuning the lifting speed. The transferred graphene exhibits a Raman 2D/G peak intensity ratio exceeding 2 (**Fig. R10a**), confirming its monolayer nature, while the absence of a D peak indicates defect-free characteristics (*Carbon* **2020**, 169, 92–98).²³ Further evidence of graphene cleanliness is provided by SEM imaging (**Fig. R10b**). (*CrystEngComm* **2021**, 23, 7406–7411).²⁴

In this experiment, the copper foil was pre-treated by electrochemical polishing. After CVD growth, the copper substrate was etched using an ammonium persulfate ((NH₄)₂S₂O₈) solution, followed by multiple rinses with deionized water to remove residual etchant. Prior to transferring graphene onto the polycrystalline diamond, the diamond substrate was ultrasonically cleaned with acetone, alcohol, and deionized water, then immersed in dilute HF solution for 5 minutes to remove surface oxides, ensuring a clean interface. Before transfer, the polycrystalline diamond surface was treated with O₂ plasma (50 W, 1 minute) to enhance hydrophilicity and promote graphene adhesion. AFM and SEM images of the graphene-diamond interface are shown in **Fig. R10c**. The polycrystalline diamond exhibited random orientations with distinct grain boundaries, and its RMS roughness was approximately 0.3 nm. Due to the influence of the polycrystalline substrate, the graphene developed varying degrees of wrinkles after wet transfer, with an RMS roughness of about 0.51 nm. The SEM images confirmed that the graphene film had no visible cracks, supporting its low-defect characteristics.

[Figure Redacted]

Figures R10 a, Raman spectra of monolayer graphene transferred by different methods.²³ **b**, SEM image of graphene before the AlN growth. AFM image of graphene before the AlN growth is shown in the inset.²⁴ **c**, SEM and AFM images of the polycrystalline diamond substrate and wet-transferred monolayer graphene.

In the revised manuscript (The second paragraph, the first sentence), the authors added: “Due to the influence of this polycrystalline substrate, the wet-transferred graphene shows varying degrees of wrinkles with an RMS roughness of about 0.51 nm. The surface morphology of both diamond and graphene was characterized by AFM and SEM, as shown in Supplementary Fig. 3.”

References:

21. Jia Y, Guo H, Ning J, et al. Flexible High-Stability Self-Variable-Voltage Monolithic Integrated System Achieved by High-Brightness LED for Information Transmission. *Small*, **17**, 2105207. (2021).
22. Chen D, Ning J, Wang D, et al. High-quality GaN grown on nitrogen-doped monolayer graphene without an intermediate layer. *Science China Materials*, **66**, 1968-1977. (2023).
23. Shen, X. et al. MMA-enabled ultraclean graphene transfer for fast-response graphene / GaN ultraviolet photodetectors. *Carbon*, **169**, 92-98 (2020).
24. Jia Y, Wu H, Zhao J, et al. Growth mechanism on graphene-regulated high-quality

epitaxy of flexible AlN film. *CrystEngComm*, **23**, 7406-7411. (2021).

Comment #2.10 Provide computational details (e.g., pseudopotentials, convergence criteria) to ensure reproducibility.

Response: We sincerely appreciate the reviewer's insightful questions regarding the computational details. The modifications and supplementary content pertaining to the computational section have been highlighted in the revised manuscript. All calculations were performed using the Adsorption Locator module in Materials Studio. The simulations were based on a forcefield approach rather than DFT, and the adsorption energies of oxygen atoms on diamond surfaces were calculated using a Monte Carlo method combined with Simulated Annealing to identify the most probable adsorption sites; therefore, the choice of pseudopotentials was not applicable. The Universal force field was employed to describe interatomic interactions, including van der Waals and electrostatic terms, with a cutoff distance of 12.5 Å for interactions. To avoid periodic boundary artifacts, a vacuum layer of 30 Å was added perpendicular to the surface. The convergence tolerances for energy and force were set to 1.0×10^{-4} kcal/mol and 0.005 kcal/mol/Å, respectively. The displacement tolerance was set to 5.0×10^{-5} Å, and the maximum number of iterations was set to 500.

In the revised manuscript (Methods, fourth paragraph), the authors revised: "All calculations were performed using the Adsorption Locator module in Materials Studio. The simulations were based on a forcefield approach, and the adsorption energies of oxygen atoms on diamond surfaces were calculated using a Monte Carlo method combined with Simulated Annealing to identify the most probable adsorption sites. The Universal force field was employed to describe interatomic interactions, including van der Waals and electrostatic terms, with a cutoff distance of 12.5 Å for interactions. To avoid periodic boundary artifacts, a vacuum layer of 30 Å was added perpendicular to the surface. The convergence tolerances for energy and force were set to 1.0×10^{-4} kcal/mol and 0.005 kcal/mol/Å, respectively. The displacement tolerance was set to 5.0×10^{-5} Å, and the maximum number of iterations was set to 500."

Comment #2.11 Shortly discuss scalability challenges (e.g., wafer-scale graphene uniformity, polycrystalline diamond grain boundaries) in terms of industrial application.

Response: Thank you for the suggestion. As you rightly pointed out, despite the promising demonstration of van der Waals β -Ga₂O₃ epitaxy on polycrystalline diamond substrates, several critical challenges must be addressed for large-scale industrial deployment. One major limitation lies in achieving wafer-scale uniformity of monolayer graphene. Current chemical vapor deposition (CVD) techniques and transfer processes often result in defects and contamination, which can compromise the interface quality and epitaxial consistency of Ga₂O₃ films.

Moreover, the polycrystalline nature of diamond substrates introduces grain boundaries with misoriented crystal domains and heterogeneous thermal conductivity. These boundaries may lead to local variations in the growth orientation, interfacial quality, and heat dissipation behavior, thus posing obstacles to the fabrication of large-area, high-performance devices. Under high-temperature operation or thermal cycling, the mismatch in thermal expansion coefficients may still lead to long-term stress accumulation and eventual failure. Although the introduction of graphene can mitigate this issue to some extent, further investigation into the reliability of large-scale integration is necessary. Overcoming these scalability bottlenecks is essential for translating the proposed approach into practical semiconductor applications.

In the revised manuscript (The thirteenth paragraph, the last sentence), the authors added: “The scalability of the proposed methodology for practical semiconductor applications is currently constrained by two key factors: The limited availability of large-area wafer-scale single-crystal diamond substrates, and challenges in maintaining monolayer graphene uniformity across full wafer dimensions. Overcoming these scalability bottlenecks represents a critical pathway for industrial implementation in future development.”

Comment #2.12 Comment on the long-term stability (e.g., interfacial degradation under high-power/frequency operation).

Response: We sincerely appreciate the reviewers for raising the critical issue of long-

term stability, and we fully agree that interfacial aging can degrade device performance in power electronics. To address this, we conducted high-resolution TEM imaging and elemental EDS mapping of the interface, as shown in **Figure R10a**. The Ga signal (blue) is predominantly localized above the graphene interlayer, corresponding to the β -Ga₂O₃ thin-film region, while being nearly undetectable below the graphene. This spatially confined elemental distribution provides direct evidence that the graphene layer effectively blocks interdiffusion, preserving interfacial chemical sharpness. The minimal C penetration is attributed to the oxygen-rich and humid mist-CVD environment during epitaxial growth, where elements may diffuse through graphene wrinkles. Furthermore, Labeled et al. demonstrated that a monolayer graphene interlayer suppresses high-temperature interfacial diffusion, thereby enhancing the thermal stability and reliability of β -Ga₂O₃ devices.²⁵ Earlier work by Morrow et al. also established graphene's effectiveness as a diffusion barrier.²⁶

We further extracted the conductivity variations of the fabricated photodetector under high optical frequencies, as shown in **Figure R11b**. The results demonstrate negligible conductivity changes with increasing light frequency, indicating efficient carrier extraction and well-passivated material defects/interface states that minimize the impact of slow recombination on high-frequency response. Although high-frequency illumination may induce localized Joule heating, the stable conductivity confirms effective thermal management in the device design, with no observable interface degradation due to temperature rise. Additionally, we measured the photocurrent response under 1 Hz optical pulse illumination over an extended duration of 2000 seconds (**Figure R11c**). The device exhibits remarkable robustness and reproducibility, with response times of $\tau_1=54$ ms (rise) and $\tau_2=4$ ms (decay). Collectively, these results demonstrate outstanding long-term stability of the photodetector.

Figure R11. **a**, Cross-sectional EDS elemental mapping of the heterointerface. **b**, Frequency-dependent conductivity response (1 Hz-1000Hz) under pulsed optical excitation. **c**, Long-term photocurrent stability test spanning 2000 s under continuous 1 Hz optical pulsing.

In the revised manuscript (The thirteenth paragraph, the first half), the authors added: “We further extracted the conductivity variations of the fabricated photodetector under high optical frequencies, as shown in Supplementary Fig. 6. The results demonstrate negligible conductivity changes with increasing light frequency, indicating efficient carrier extraction and well-passivated material defects/interface states that minimize the impact of slow recombination on high-frequency response. Although high-frequency illumination may induce localized Joule heating, the stable conductivity confirms effective thermal management in the device design, with no observable interface degradation due to temperature rise. Additionally, we measured the photocurrent response under 1 Hz optical pulse illumination over an extended duration of 2000 seconds. The device exhibits remarkable robustness and reproducibility, with response times of $\tau_1=54$ ms (rise) and $\tau_2=4$ ms (decay). Collectively, these results demonstrate outstanding long-term stability of the photodetector.”

References:

25. Labeled et al. Transferred Graphene Monolayer to β -Ga₂O₃ as a Diffusion Barrier for Based Power Device Applications. *ACS nano*, **19**, 8842-8851 (2025).
26. Morrow, W. K., Pearton, S. J., & Ren, F. Review of graphene as a solid state diffusion barrier. *Small*, **12**, 120-134 (2016).

Comment #2.13 Is the dark current value of $5.83 \times 10^{-6} \mu\text{A}$ not rather low?

Response: We appreciate the reviewer's question regarding our dark current data. In our photodetector, the measured dark current is $5.83 \times 10^{-6} \mu\text{A}$, equivalent to 5.83×10^{-12} A. Based on our survey of relevant literature, dark current values at this level are quite common. For instance, as early as 2018, Yaxuan Liu et al. reported a Ga₂O₃ solar-blind photodetector with a dark current as low as 5 pA (5×10^{-12} A)²⁷. In 2019, Yuan Qin et al. developed a photodetector based on β -Ga₂O₃ with a dark current as low as 0.7 pA (7×10^{-13} A)²⁸. In 2021, Zeng Liu et al. reported a β -Ga₂O₃ field-effect deep-ultraviolet phototransistor with a dark current of 13.4 pA (1.34×10^{-11} A)²⁹. Also in 2019, Xiaohu Hou et al. fabricated a solar-blind photodetector based on α -phase-dominated Ga₂O₃ films with a dark current as low as 81 fA (8.1×10^{-14} A)³⁰. Recently in 2024, Hu, Z. et al. fabricated a photodetector based on Al-doped Ga₂O₃, achieving an ultralow dark current of 156 fA (1.56×10^{-13} A), thereby realizing an ultrahigh photodetection contrast ratio (PDCR).³¹ Numerous similar reports exist, indicating that the dark current value obtained in this study is within a common range.

27. Liu, Y. et al. Ga₂O₃ Field-Effect-Transistor-Based Solar-Blind Photodetector With Fast Response and High Photo-to-Dark Current Ratio. *IEEE Electron Device Letters*, **39**, 1696-1699 (2018).
28. Qin, Y., et al. Enhancement-Mode β -Ga₂O₃ Metal–Oxide–Semiconductor Field-Effect Solar-Blind Phototransistor With Ultrahigh Detectivity and Photo-to-Dark Current Ratio. *IEEE Electron Device Letters*, **40**, 742-745 (2019).
29. Liu, Z., et al. Enhancement-mode normally-off β -Ga₂O₃: Si metal-semiconductor field-effect deep-ultraviolet phototransistor. *Semiconductor Science and Technology*, **37**, 015001 (2021).
30. Hou, X., et al. Ultrahigh-Performance Solar-Blind Photodetector Based on α -Phase-

Dominated Ga₂O₃ Film With Record Low Dark Current of 81 fA. *IEEE Electron Device Letters*, **40**, 1483-1486 (2019).

31. Hu, Z. P. et al. Ultra-high PDCR (> 109) of vacuum-UV photodetector based on Al-doped Ga₂O₃ microbelts. *Nanotechnology*, **36**, 025202. (2024).

Response to the Reviewer 3

Overall Assessment from Reviewer 3:

Response: We truly appreciate the reviewer's involvement in the review process as part of the Nature Communications initiative. We are honored that our work contributed to the peer review training of Early Career Researchers.

Response to the Reviewer 4

Overall Assessment from Reviewer 4:

This manuscript presents an excellent study that demonstrates the interface stress relaxation, high crystallinity, enhanced thermal conductivity, and high-performance UV photodetection characteristics of β-Ga₂O₃ thin films. In particular, the authors clearly recognize the limitations of wafer-scale single-crystal diamond substrates and the structural defects associated with conventional heteroepitaxial approaches. Their adoption of the Van der Waals epitaxy (VdW epitaxy) strategy to overcome these issues is both innovative and practically meaningful.

Response: We greatly appreciate the reviewer's recognition and encouraging feedback. In the revised version, we have carefully addressed all comments raised by the reviewers. We are also grateful for the constructive suggestions, which have helped us further improve the clarity and overall quality of the manuscript.

Comment #4.1 What are the specific advantages of mist-CVD-based VdW epitaxy compared to MOCVD, HVPE, or bonding methods? A comparative explanation, particularly regarding nucleation on chemically inert surfaces like graphene, would be helpful.

Response: We appreciate the reviewer’s insightful comments regarding the discussion of the mist-CVD method. Compared with the aforementioned techniques, mist-CVD is a non-vacuum-based approach, which significantly reduces the energy consumption typically required to maintain vacuum conditions. The equipment setup is relatively simple, requiring only atomization and heating systems to operate. Furthermore, mist-CVD offers distinct advantages such as diverse selection of growth precursors, enhanced safety, and low cost. For example, the precursors are inexpensive metal salts, eliminating the need for costly organometallic compounds or high-purity gases. The method generates no toxic gases, making it highly safe and well-suited for both laboratory research and industrial-scale production. It also exhibits scalability potential, allowing for large-area deposition without the need for the complex gas flow designs in MOCVD or the vacuum chamber constraints in MBE.

Table R1. Comparison of characteristics of common epitaxial growth methods

	Mist-CVD	MOCVD	MBE	Magnetron Sputtering	ALD
Growth Rate	Medium (10-100 nm/min)	High (0.1-10 $\mu\text{m}/\text{h}$)	Very Low (0.01-1 $\mu\text{m}/\text{h}$)	Medium (10-500 nm/min)	Very Low (0.1-0.3 nm/cycle)
Uniformity	Large-Area Uniform	Large-Area Uniform	Small-Area Uniform	Large-Area Uniform	Extremely Uniform
Temperature Range	RT - 800°C	600 - 1200°C	200 - 800°C	RT - 600°C	RT - 400°C
Vacuum Requirement	Ambient/Low Pressure	Low Pressure	Ultra-High Vacuum	Medium Vacuum	Low Pressure
Compatible Materials	Oxides (ZnO, Ga ₂ O ₃)	III-V/II-VI Semiconductors	Semiconductors/Superlattices	Metals/Oxides/Nitrides	Oxides/Nitrides/Metals
Equipment Cost	Low	Very High	Very High	Medium	High
Precursor Cost	Low (Aqueous)	High (Toxic Gases)	High (Ultra-Pure Solids)	Medium (Targets)	Medium-High

Environmental Impact	High (Non-Toxic)	Low (Toxic Emissions)	Medium (Vacuum)	Medium (Inert Gas)	Medium (Precursors)
Unique Advantages	Low-Temp, Low-Cost	High-Speed, High-Throughput	Atomic Precision	Broad Material Compatibility	3D Conformal Coverage

However, it is worth noting that further improvements are still needed in film uniformity. In contrast, wafer bonding relies heavily on the properties of the existing material surface, which may introduce interfacial defects or stress, and the high temperatures involved could lead to the formation of amorphous interfacial layers.

In addition, mist-CVD presents unique advantages for nucleation and growth on chemically inert surfaces such as graphene and h-BN, particularly in terms of low-temperature activation, precursor adaptability, and uniform nucleation. Mist-CVD enables the use of oxygen- or nitrogen-containing precursors (e.g., zinc nitrate, gallium acetylacetonate), which partially decompose during atomization to produce reactive species such as –OH or –COOH. These species locally modify the graphene surface and promote nucleation. In contrast, MOCVD or MBE requires high temperatures (>600 °C) or plasma assistance to activate graphene. Micron-sized atomized droplets temporarily reside on the graphene surface and enhance adsorption via van der Waals forces or capillary effects, whereas vapor-phase molecules in conventional deposition methods are more prone to desorption.

In the revised manuscript (The third paragraph, the second half), the authors added: “This non-vacuum method reduces energy costs via simple atomization/heating systems, using affordable, safe precursors without toxic byproducts. It enables scalable, large-area deposition and excels at low-temperature nucleation on inert surfaces (e.g., graphene/h-BN) through reactive intermediates (-OH/-COOH) from partially decomposed precursors, unlike MOCVD/MBE’s high-temperature/plasma requirements. Droplet-enhanced adsorption further aids uniform growth. Mist-CVD avoids wafer bonding’s interfacial defects/stress or amorphous layer risks. These advantages suit both lab and industrial use.”

Comment #4.2 In the DFT calculations, does the use of a single-crystal vs.

polycrystalline substrate significantly affect the calculated binding energy? Ensuring consistency with the actual experimental conditions is important for reliable interpretation.

Response: We sincerely appreciate the reviewer's thoughtful concerns regarding the computational methodology. We fully acknowledge that maintaining consistency between computational and experimental conditions is crucial for reliable interpretation. The full-scale simulation of polycrystalline substrates is computationally intractable and cannot perfectly replicate the truly random orientations in reality, our single-crystal model remains representative.

As evidenced by the substrate XRD peaks in Manuscript Figure 3a, the (111) orientation dominates in our polycrystalline diamond substrate, which aligns with the crystal plane selected for our model. Although polycrystalline substrates consist of randomly oriented grains, the interfacial adsorption energy is primarily determined by local atomic arrangements. **(Fig. R12a)** Our experimental characterization confirms that most exposed grain surfaces match the model configuration, validating the representativeness of our calculations. **(Fig. R12b)** Furthermore, our computational predictions demonstrate that monolayer graphene/BN can significantly enhance adsorption energy (Figure 1b), with improvements observed across various crystal orientations. This confirms that despite grain randomness, the overall adsorption behavior can be effectively predicted by single-crystal modeling. We have highlighted these clarifying points in the revised manuscript.

Figure R12. a, Surface morphology SEM image of polycrystalline diamond. **b,** Atomic schematic diagram of diamond with different orientations.

In the revised manuscript (The second to last sentence of Paragraph four), the authors added: “Although polycrystalline substrates consist of randomly oriented grains, the interfacial adsorption energy is primarily determined by local atomic arrangements. Our experimental characterization confirms that most exposed grain surfaces match the model configuration, validating the representativeness of calculations. Furthermore, our computational predictions demonstrate that monolayer graphene/BN can significantly enhance adsorption energy, with improvements observed across various crystal orientations.”

Comment #4.3 Could the uniformity of the graphene transfer process—including potential defects, contamination, or mechanical strain—have influenced the crystallinity and device performance of the resulting Ga₂O₃ films? In particular, please discuss the correlation with grain boundary formation, such as those observed at 760 °C in Fig. 4(d).

Response: We sincerely appreciate the reviewer's insightful questions. Indeed, defects potentially introduced during graphene transfer - such as cracks, wrinkles, polymer residues, or tensile stress - may affect the nucleation density and growth orientation of Ga₂O₃. Our research group has extensive expertise in graphene transfer technology. Benefiting from ultraclean interfaces, we have successfully achieved stress-free nitride film epitaxy (*Applied Sciences*, **2020**, 10, 8814)²⁷, enabling high-brightness violet light-emitting diodes (*Advanced Optical Materials*, **2020**, 8, 1901632)²⁸ and light-emitting diode applications (*ACS Applied Materials & Interfaces*, **2021**, 13, 32442-32449).²⁹ Previous studies have also demonstrated the feasibility of obtaining ultraclean graphene (*Nano Letters*, **2012**, 12, 414-419).³⁰ Guided by these methodologies and experiences, we have successfully realized nearly defect-free monolayer graphene, with its surface characteristics demonstrated by SEM and AFM images in **Figure R13a**.

Figure R13. a, SEM and AFM images of the polycrystalline diamond substrate and wet-transferred monolayer graphene on polycrystalline diamond. **b**, SEM images of β -Ga₂O₃ before and after van der Waals epitaxy on graphene/poly-diamond and grown under O₂ flow rates of 300 and 600 sccm.

Furthermore, heterogeneous nucleation frequently leads to misoriented grains that coalesce into grain boundaries during film growth. As shown in Figure 4(d), distinct grain boundaries are observed at the growth temperature of 760°C, which may originate from: (i) defects in the transferred graphene, (ii) post-growth thermal stress relaxation, or (iii) gas flow rate effects.

Figure R13b presents SEM images of Ga₂O₃ grown on graphene/polycrystalline diamond substrates, revealing that graphene surface wrinkles induce local growth rate variations, manifested as non-uniform image contrast. Similarly, excessive gas flow rates can prematurely sweep away precursor molecules before complete surface adsorption occurs, resulting in unstable deposition rates and spatial inhomogeneity. Conversely, insufficient flow rates (e.g., 300 sccm in our experiments) cause inadequate precursor supply and excessively slow growth, yielding films with high surface roughness (RMS=51.3 nm). These findings underscore the necessity for careful optimization of gas flow parameters to ensure film quality in practical applications.

References:

27. Zeng Y, Ning J, Zhang J, et al. Raman analysis of E2 (high) and A1 (LO) phonon to the stress-free GaN grown on sputtered AlN/graphene buffer layer. *Applied Sciences*, **10**, 8814. (2020).

28. Jia Y, Ning J, Zhang J, et al. Transferable GaN enabled by selective nucleation of AlN on graphene for high-brightness violet light-emitting diodes. *Advanced Optical Materials*, **8**, 1901632. (2020).
29. Jia Y, Ning J, Zhang J, et al. High-quality transferred GaN-based light-emitting diodes through oxygen-assisted plasma patterning of graphene. *ACS Applied Materials & Interfaces*, **13**, 32442-32449. (2021).
30. Lin, Y. C., Lu, C. C., Yeh, C. H., Jin, C., Suenaga, K., & Chiu, P. W. Graphene annealing: how clean can it be? *Nano letters*, **12**, 414-419. (2012).

Comment #4.4 Is the reported effective thermal conductivity value (95.8 W/m·K) derived with consideration of measurement uncertainty, contact resistance, and boundary conditions? A more detailed explanation would improve the reliability of the thermal analysis.

Response: We thank the reviewers for their valuable questions regarding the effective thermal conductivity of thin films. Initially, we attempted to measure the samples using laser thermal conductivity analysis, but this method's low spatial resolution made it unsuitable for measuring ultrathin films (<1 μm) or nanoscale interfaces, introducing significant errors. The time-domain thermoreflectance (TDTR) technique based on ultrafast lasers has emerged as a reliable, powerful, and versatile method (*Journal of Applied Physics*, **2018**, 124)³¹ for measuring thermal properties of various thin film materials and their interfaces (*ACS Applied Electronic Materials*, **2024**, 6, 1710-1717),³² which we adopted to overcome these limitations and obtain accurate measurements.

To address this issue and the reviewers' concerns, we have supplemented the thermal conductivity measurements of Ga₂O₃ using time-domain thermoreflectance (TDTR), along with the Ga₂O₃/diamond interfacial thermal resistance for different samples, as shown in **Fig. R14c** and **d**. Here, three test samples were coated with 100 nm Al films via electron-beam evaporation, with a pump spot radius of 12 μm (at $1/e^2$ intensity) and a probe spot radius of 4 μm (at $1/e^2$ intensity). The measurements were conducted at a modulation frequency of 1 MHz under 25 °C ambient conditions. By

jointly fitting the experimental curves with theoretical models, we obtained the thermal conductivity parameters of the gallium oxide samples and the interfacial thermal resistance values at the Ga₂O₃-diamond interface. The directly epitaxial sample exhibited the lowest effective thermal boundary resistance (TBR_{eff}) of 2.82 m²·K/GW and the highest thermal conductivity of 7.59 W/(m·K).

[Figure Redacted]

[Figure Redacted]

Figure R14. **a**, Schematic diagram of the sample structure and thermoreflectance measurement scheme.³² **b**, Schematic of a typical transient thermoreflectance setup.³¹ **c**, Thermal conductivity of Ga₂O₃ films and interfacial thermal boundary resistance (TBR_{eff}) for samples. **d**, Benchmark comparison of TBR_{eff} values across different interface engineering approaches.

In the revised manuscript (Methods, the third Paragraph), the authors added: “Supplemented our study with time-domain thermoreflectance (TDTR, Pioneer-ONE) measurements of both the thermal conductivity and thermal boundary resistance. For these tests, we deposited 100 nm Al films on all three samples via e-beam evaporation, with a pump beam spot radius of 12 μm (at 1/e² intensity) and a probe beam spot radius of 4 μm (at 1/e² intensity), using a 1 MHz modulation frequency at 25°C. By fitting the

measurement curves with theoretical models, we obtained the Ga₂O₃ thermal conductivity parameters and Ga₂O₃-diamond thermal boundary resistance values.”

References:

31. Jiang, P., Qian, X., & Yang, R. Tutorial: Time-domain thermoreflectance (TDTR) for thermal property characterization of bulk and thin film materials. *Journal of Applied Physics*, **124**. (2018).
32. Xu, W., et al. Thermal transport properties of β -Ga₂O₃ thin films on Si and SiC substrates fabricated by an ion-cutting process. *ACS Applied Electronic Materials*, **6**, 1710-1717. (2024).

Comment #4.5 How might the variation in film thickness affect crystallinity and the measured effective thermal conductivity? Especially for thin films, could the high thermal conductivity of the underlying substrate lead to an overestimation of the intrinsic thermal properties of β -Ga₂O₃?

Response: We sincerely appreciate the reviewers' insightful comments regarding the effective thermal conductivity of thin films. We have addressed these concerns in the revised manuscript with yellow highlights for easy identification. As rightly pointed out, film thickness variations significantly impact both crystallinity and thermal conductivity. Thinner films (<100 nm) often exhibit reduced crystalline integrity due to interfacial strain or amorphization effects, while thicker films (>500 nm) typically demonstrate more uniform grain growth and lower defect densities. This thickness dependence was notably demonstrated in 2019 when researchers revealed that the thermal conductivity of specifically-oriented Ga₂O₃ increases with thickness (*APL Materials*, **2019**, 7, 031118).³³ Furthermore, the well-documented thickness-dependent thermal conductivity (*Applied Physics Letters*, **2020**, 116)³⁴ follows the Callaway model, where enhanced phonon boundary scattering in thinner films leads to significantly reduced effective thermal conductivity.

To eliminate potential substrate interference in thermal measurements, we employed time-domain thermoreflectance (TDTR) for direct assessment of the thin film's in-plane thermal conductivity. This technique effectively circumvents substrate

effects through time-resolved analysis and lateral heat flow characterization. Specifically, we analyzed signals in the early time window (<1 ns) when thermal energy remains confined within the film before significant substrate diffusion occurs, thereby exclusively reflecting the film's intrinsic thermal transport properties. (*Journal of Applied Physics*, **2018**, 124.)³⁵ Through Fourier transform analysis of the thermoreflectance signals, we extracted high-frequency components (>1 MHz) that correspond to rapid thermal responses predominantly governed by the film characteristics. (*Nature*, **2021**, 597, 660-665.)³⁶ For our 350 nm-thick β -Ga₂O₃ film exhibiting optimal crystallinity with a (-201) plane rocking curve FWHM of 0.18°, the measured thermal conductivity reached 7.19 W/(m·K).

[Figure Redacted]

Figure R15. a, Thickness dependent thermal conductivity of Ga₂O₃ thin films.³³ **b**, Thickness-dependent thermal conductivity of β -Ga₂O₃ thin films from both TDTR measurements.³⁴

In the revised manuscript (The eleventh Paragraph), the authors added: “As demonstrated by Zheng et al., the thermal conductivity of Ga₂O₃ with specific orientations increases with thickness.¹¹ In our experiments, the ($\bar{2}01$) plane rocking curve showed a minimum FWHM of 0.18° for the 350-nm-thick film, corresponding to a thermal conductivity of 7.19 W/m·K. Furthermore, thinner films display more pronounced phonon boundary scattering (consistent with the Callaway model), leading to reduced effective thermal conductivity.”

References:

33. Cheng, Z., Yates, L., Shi, J., Tadjer, M. J., Hobart, K. D., & Graham, S. Thermal conductance across β -Ga₂O₃-diamond van der Waals heterogeneous interfaces. *Apl*

- Materials*, **7**, 031118 (2019).
34. Zhang, Y., Su, Q., Zhu, J., Koirala, S., Koester, S. J., & Wang, X. Thickness-dependent thermal conductivity of mechanically exfoliated β -Ga₂O₃ thin films. *Applied Physics Letters*, **116**. (2020).
 35. Jiang, P., Qian, X., & Yang, R. Tutorial: Time-domain thermoreflectance (TDTR) for thermal property characterization of bulk and thin film materials. *Journal of Applied Physics*, **124**. (2018).
 36. Kim, S. E. et al. Extremely anisotropic van der Waals thermal conductors. *Nature*, **597**, 660-665. (2021).